# NuTime: Numerically Multi-Scaled Embedding for Large-Scale Time-Series Pretraining

**Chenguo Lin**[*][†]                                                     *chenguolin@stu.pku.edu.cn*
*Peking University*

**Xumeng Wen**[†]                                                        *xumengwen@microsoft.com*
*Microsoft Corporation*

**Wei Cao**                                                                 *weicao@microsoft.com*
*Microsoft Corporation*

**Congrui Huang**                                                        *conhua@microsoft.com*
*Microsoft Corporation*

**Jiang Bian**                                                          *jiang.bian@microsoft.com*
*Microsoft Corporation*

**Stephen Lin**                                                         *stevelin@microsoft.com*
*Microsoft Corporation*

**Zhirong Wu**                                                         *wuzhiron@microsoft.com*
*Microsoft Corporation*

**Reviewed on OpenReview:** *https://openreview.net/forum?id=TwiSBZOp9u*

## Abstract

Recent research on time-series self-supervised models shows great promise in learning semantic representations. However, it has been limited to small-scale datasets, e.g., thousands of temporal sequences. In this work, we make key technical contributions that are tailored to the numerical properties of time-series data and allow the model to scale to large datasets, e.g., millions of temporal sequences. We adopt the Transformer architecture by first partitioning the input into non-overlapping windows. Each window is then characterized by its normalized shape and two scalar values denoting the mean and standard deviation within each window. To embed scalar values that may possess arbitrary numerical amplitudes in a high-dimensional space, we propose a numerically multi-scaled embedding module enumerating all possible numerical scales for the scalars. The model undergoes pretraining with a simple contrastive objective on a large-scale dataset over a million sequences collected by merging existing public data. We study its transfer performance on a number of univariate and multivariate classification tasks, few shot learning, unsupervised clustering and anomaly detection benchmarks. Our method exhibits remarkable improvement against previous pretraining approaches and establishes the new state of the art, even compared with domain-specific non-learning-based methods. Code is available at: *https://github.com/chenguolin/NuTime*.

---

[*]This work was done during an internship at Microsoft. † indicates equal first author contribution.

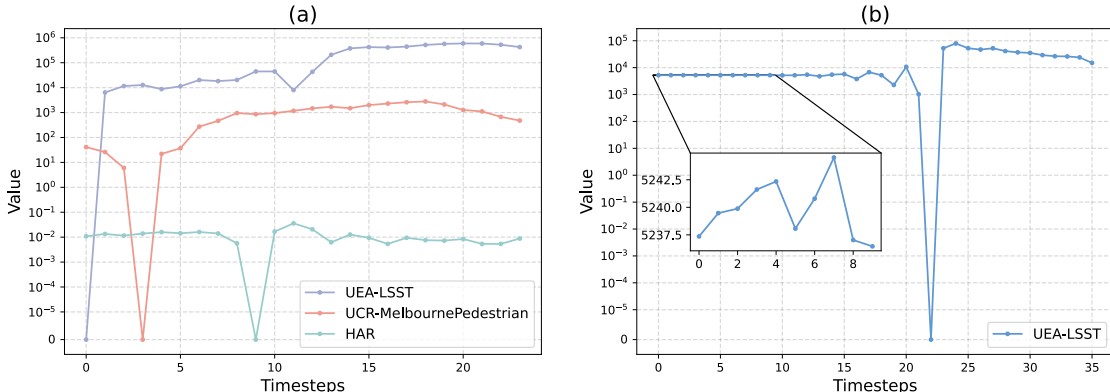

Figure 1: (a) Numerical scales of three temporal sequences from three datasets differ significantly. (b) Even a single sequence may contain multiple scales of numerical variations. The zoom-in view shows the local structure of small variations. Note that sequences are shifted above the x-axis and presented in a logarithmic scale for better visualizations.

## 1 Introduction

Despite the phenomenal achievement of large-scale representation learning on various data modalities (Brown et al., 2020; Radford et al., 2021; Caron et al., 2021), research for time-series representation learning is mostly limited to small-scale datasets without attaining generalization capabilities (Eldele et al., 2021b; Yue et al., 2022; Zhang et al., 2022). Since time-series data may cover a diverse range of domains, such as medicine, weather and traffic, large-scale training across domains brings special challenges for transfer learning.

We notice a unique characteristic of time-series data and its representation. While RGB images are represented by fixed and discretized numerical values from 0 to 255, and natural languages are tokenized to a fixed dictionary, time-series data exhibit numerical values that are continuous and of drastically different amplitudes. For instance, temperature usually varies from -30 to 30 degrees Celsius, but altitude is on the scale of $10^3$ meters. The scale of numerical variations generally depends on the physical properties of the time-series data. As illustrated in Figure 1, sequences from certain time-series categories and even a single time-series sequence may exhibit structures at multiple scales due to a change in physical properties.

Deep neural networks trained with gradient descent need proper normalization for optimization to a good local minimum (Ioffe & Szegedy, 2015; Ba et al., 2016). However, encoding time-series data to a normalized vector space is a non-trivial problem. Z-score normalization is a popular technique that assumes a single dominant numerical scale in the dataset. Instance normalization[1] preprocesses data by per-instance statistics, and it thus removes information that is critical for representation learning. As a result, both of the conventional data encoding methods fail to effectively encode time-series data with a high variation of numerical scales (i.e., amplitude). The dilemma between normalization for effective network optimization and high variation of data scales poses a challenge for time-series representation learning, especially in the large-scale scenario.

We introduce **NuTime**, a Transformer-based architecture for time-series data with a novel embedding module to effectively embed data of arbitrary numerical amplitude. For a time-series sequence, we first divide it into non-overlapping small windows, so that data within each window has a simple structure that can be easily modeled at a single scale. A window is characterized by three factors: its mean and standard deviation (std), and the normalized shape. Embedding vectors of the three factors are combined and fed as an input token to a general-purpose Transformer for representation learning. The normalized shape across both windows and samples are of similar numerical scales, and thus can be embedded by a simple linear layer. The challenge of embedding the entire sequence reduces to the embedding of a set of means and standard deviations, whose numerical amplitudes may vary arbitrarily.

---

[1] Instance normalization in this paper refers to normalizing each sample individually, which is non-parametric and different from the process proposed by Ulyanov et al. (2016).

To encode these scalars to a high-dimensional vector space, we propose a Numerically Multi-scaled Embedding (**NME**) module. Since encoding through network modules may need to assume the numerical scale of inputs (Ioffe & Szegedy, 2015; Ba et al., 2016; Wu & He, 2018), our idea is to simply enumerate all possible scales for arbitrary input scalar and later fuse these embeddings across numerical scales. We use a basic building block of a linear layer followed by a layer normalization (Ba et al., 2016) to map a scalar to a normalized vector. Such a basic building block is sensitive to the input range, which we find is controlled by a multiplier on the bias term in the linear layer. We thus use paralleled building blocks with different bias multipliers set for each numerical scale. The output embeddings are aggregated by a weighting mechanism to derive the final scalar embedding.

To conduct large-scale representation learning, we collect pretraining data by combining existing datasets from various sources, yielding a dataset with over one million time-series sequences. We pretrain our NuTime model using a straightforward Bootstrap Your Own Latent (BYOL) (Grill et al., 2020) self-supervised learning objective and study the transfer performance on popular classification benchmarks. NuTime achieves the new state-of-the-arts results across univariate and multivariate time series classification tasks, and it is the first time deep learning models beats traditional approaches (Middlehurst et al., 2021b). We also demonstrate that NuTime outperforms recent approaches on few-shot learning without being designed for this task. Moreover, it can easily transfer to other downstream tasks, such as clustering and anomaly detection.

In summary, this work makes three key contributions:

- We propose a numerically multi-scaled embedding module for encoding scalar values in a wide range into a normalized vector space, which allows time series models to scale to large datasets.

- We design a general-purpose Transformer-based solution for time series representation learning with each input token representing its shape embedding, mean embedding, and std embedding.

- We conduct large-scale self-supervised pretraining for time-series data and demonstrate that transferable representations could be learned from the vast set of disparate data. It is the first time that deep learning models outperform traditional methods on the UCR and UEA classification archives.

## 2 Related Work

### 2.1 Deep Learning for Time-Series Analysis

The tasks of interest for time-series analysis include classification, anomaly detection, and forecasting. While many efforts have applied deep learning to time series, a consensus has not been reached on what is the most suitable neural architecture, and in what circumstances can deep learning outperform a traditional approach. For classification, some of the leading methods are based on convolutional inception networks (Ismail Fawaz et al., 2020) and recurrent neural networks (Che et al., 2018). For forecasting, LSTMs (Shih et al., 2019), graph neural networks (Wu et al., 2020), Transformers (Zhou et al., 2021; Wu et al., 2021; Nie et al., 2023) and even pure MLP-based networks (Oreshkin et al., 2019) all achieve competitive performance. Our work does not deal with designing the backbone architecture for time series, but rather it focuses on encoding numerical values of different scales of variation.

### 2.2 Unsupervised Representation Learning for Time Series

Several studies have successfully applied unsupervised representation learning to time-series data. T-Loss (Franceschi et al., 2019) is a leading effort that combines a dilated causal architecture and a triplet loss. TNC (Tonekaboni et al., 2021), TS-TCC (Eldele et al., 2021a) and TS2Vec (Yue et al., 2022) further incorporate dedicated learning objectives, e.g., contextual and hierarchical losses, and handcrafted augmentation functions. TST (Zerveas et al., 2021), Ti-MAE (Li et al., 2023) and SimMTM (Dong et al., 2023) formulate masked modeling frameworks for time series representation learning. BTSF (Yang & Hong, 2022) and TF-C (Zhang et al., 2022) introduce a complementary frequency domain with consistency between the temporal and the frequency domain as the supervision signal. All of these works show that the unsupervised pretrained model can offer a substantial improvement against the fully supervised counterpart.

Albeit encouraging, most of these works limit pretraining to small datasets and focus on a "one-to-one" scenario, i.e., pretrain on a single dataset and finetune on the same or a similar domain. Zhang et al. (2022) take a step further and investigate a "one-to-many" setting that finetunes an EEG-pretrained model for either hand-gesture recognition or mechanical fault prediction. They also discuss a "many-to-one" setting where the model is pretrained on a mixture of multiple datasets and subsequently finetuned on a single pure dataset. However, the finetuning performance decreases with increasing heterogeneity of pretraining datasets. The essence of unsupervised pretraining, which is capable of taking advantage of large amounts of data, remains to be explored.

### 2.3 Numerical Data Normalization

Data encoding and normalization play a key role in machine learning systems. Z-score and instance normalization are two popular methods commonly used in time-series analysis. Z-score assumes a single numerical data scale and normalizes the data according to dataset statistics. Instance normalization standardizes each sample to zero mean and unit standard deviation, but removes part of the information to recover the raw data. To address this issue, reversible instance normalization (Kim et al., 2021) is proposed to add back the mean and std statistics at the network predictions for forecasting problems. Gorishniy et al. (2022) explores embedding numerical features using piece-wise linear and periodic functions for tabular data. Morel et al. (2022) attempts to formulate multi-scaled invariant features along the temporal dimension. Our work is similar in proposing an effective numerical embedding for scalar values. However, the goal of this work is to enable large-scale pretraining for time-series data.

## 3 NuTime

### 3.1 Problem Statement

We aim to conduct self-supervised pretraining for time series on a large-scale dataset, covering various domains with very different signal characteristics. Due to the nature of the physical properties, time-series data may have different numerical scales of variations. For example, sequences belonging to a certain category may exhibit variations on a numerical scale of 0.01, whereas those from another category may vary on a numerical scale of 10000. Variation scales may even change within a single time-series sequence. Joint training on such diverse large-scale datasets introduces new challenges.

Normalization for data preprocessing is a viable technique for mitigating the aforementioned issue. Popular normalization methods include (1) Z-score and (2) instance normalization. Z-score involves computing the mean and standard deviation statistics for the entire dataset, and normalizing each sample by subtracting the mean and dividing by the std. However, it does not address the numerical challenge for the following reasons:

- Z-score assumes a single domain scale of the dataset. Samples of this scale may be properly normalized, while samples out of the scale may be badly normalized.
- During transfer learning, the target domain may not share the same statistics as those from the training dataset, and thus suffer from this domain mismatch.

Instance normalization operates by standardizing each sample using its respective per-sample mean and standard deviation. After processing, each sample is guaranteed to have zero mean and unit standard deviation. The caveats for this method include:

- Essential information about the statistics of samples is removed, which could be detrimental to representation learning.
- Instance normalization assumes a single scale of variation within a single sample. It will be ineffective if the sequence is long and contains multiple scales of variations.

Based on these observations, we propose our approach for modeling large-scale time-series data.

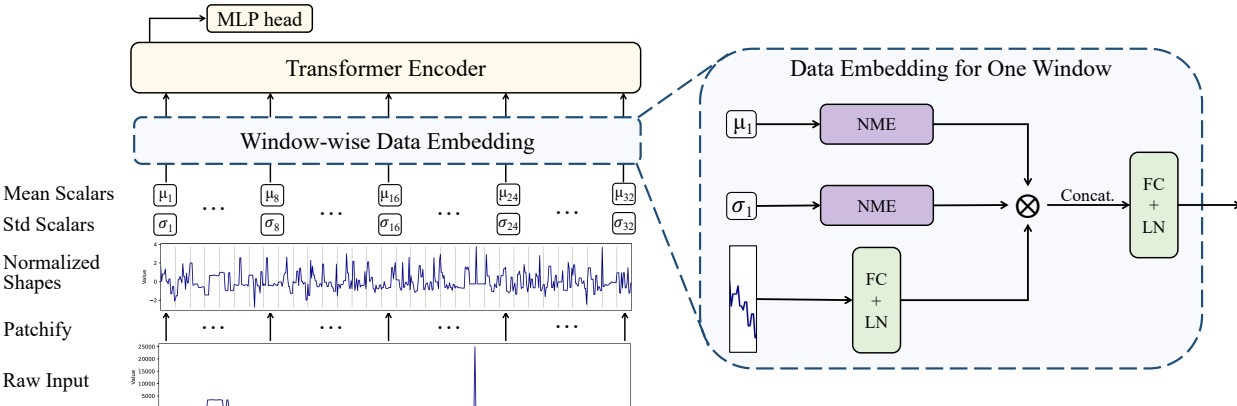

Figure 2: **Architecture Overview**. The proposed model first patchifies the input sequence into non-overlapping windows. Each window is represented by its normalized shape, the window mean and the window std. Embeddings for the three components are concatenated and transformed to be fed as input tokens into a Transformer encoder. Details about the numerically multi-scaled embedding (NME) for scalar values are explained in Section 3.3.

## 3.2 Architecture Overview

To build the pretraining model for time-series analysis, we exploit the general-purpose Transformer (Vaswani et al., 2017; Nie et al., 2023), as it has been successfully adopted in natural language, vision and time-series. The idea is to convert the input sequence into a set of tokens and then feed the tokens into the Transformer architecture. An extra `[CLS]` token is inserted at position 0 in the input, and an MLP head is appended after the Transformer encoder for classification. In the following, we will assume the time-series data to be univariate for simplicity. The extension to the multivariate case is explained in Section 3.4.

The overall architecture for NuTime is depicted in Figure 2. Each time-series sequence is split into non-overlapping windows in a similar fashion to non-overlapping patches used in Vision Transformers Dosovitskiy et al. (2020). Given the numerical challenges described earlier, it is not feasible to embed each window using a simple linear layer. Instead, we may assume that data within each window has a single scale of variation given the window size is sufficiently small. We normalize each window by its mean and std, then represent the window by three factors: the normalized shape, the mean scalar, and the std scalar. We concatenate the normalized shape embedding and the two scalar embeddings, and further project the concatenated one to the feature dimension of the Transformer. The resultant embedding is treated as an input token. After being added with a positional encoding, it is fed to the Transformer encoder.

The normalized shape can be easily embedded with a linear layer and a layer normalization. However, embedding scalar values of unknown scales of variations is less obvious.

## 3.3 Numerically Multi-scaled Embedding

**A Case Study of Linear + LayerNorm for Encoding Scalars** We begin the study by analyzing and understanding the encoding behavior of a simple linear layer followed by layer normalization (LayerNorm) (Ba et al., 2016). LayerNorm is crucial here as the linear layer maintains the magnitude of the input scalar, which would cause unstable optimization for neural network training. Denoting the input scalar as $x$, we can express this simple encoding block as follows:

$$\mathbf{z} = \mathrm{FC}(x) = x \cdot \mathbf{w} + k \cdot \mathbf{b} \quad (k = 1 \text{ by default}), \tag{1}$$

$$\mathbf{y} = \mathrm{LN}(\mathbf{z}) = \boldsymbol{\gamma} * \frac{\mathbf{z} - \mathrm{E}[\mathbf{z}]}{\sqrt{\mathrm{Var}[\mathbf{z}]}} + \boldsymbol{\beta} = \boldsymbol{\gamma} * \frac{x \cdot \mathbf{w} + k \cdot \mathbf{b}}{\sqrt{x^2 \sigma_{\mathbf{w}}^2 + k^2 \sigma_{\mathbf{b}}^2}} + \boldsymbol{\beta}, \tag{2}$$

where $\mathbf{w}$ and $\mathbf{b}$ are parameters of the linear layer, randomly initialized from the Gaussian distribution $\mathcal{N}(0, \sigma_{\mathbf{w}}^2)$ and $\mathcal{N}(0, \sigma_{\mathbf{b}}^2)$ respectively. $\boldsymbol{\gamma}$ and $\boldsymbol{\beta}$ are learnable affine parameters for the layer normalization

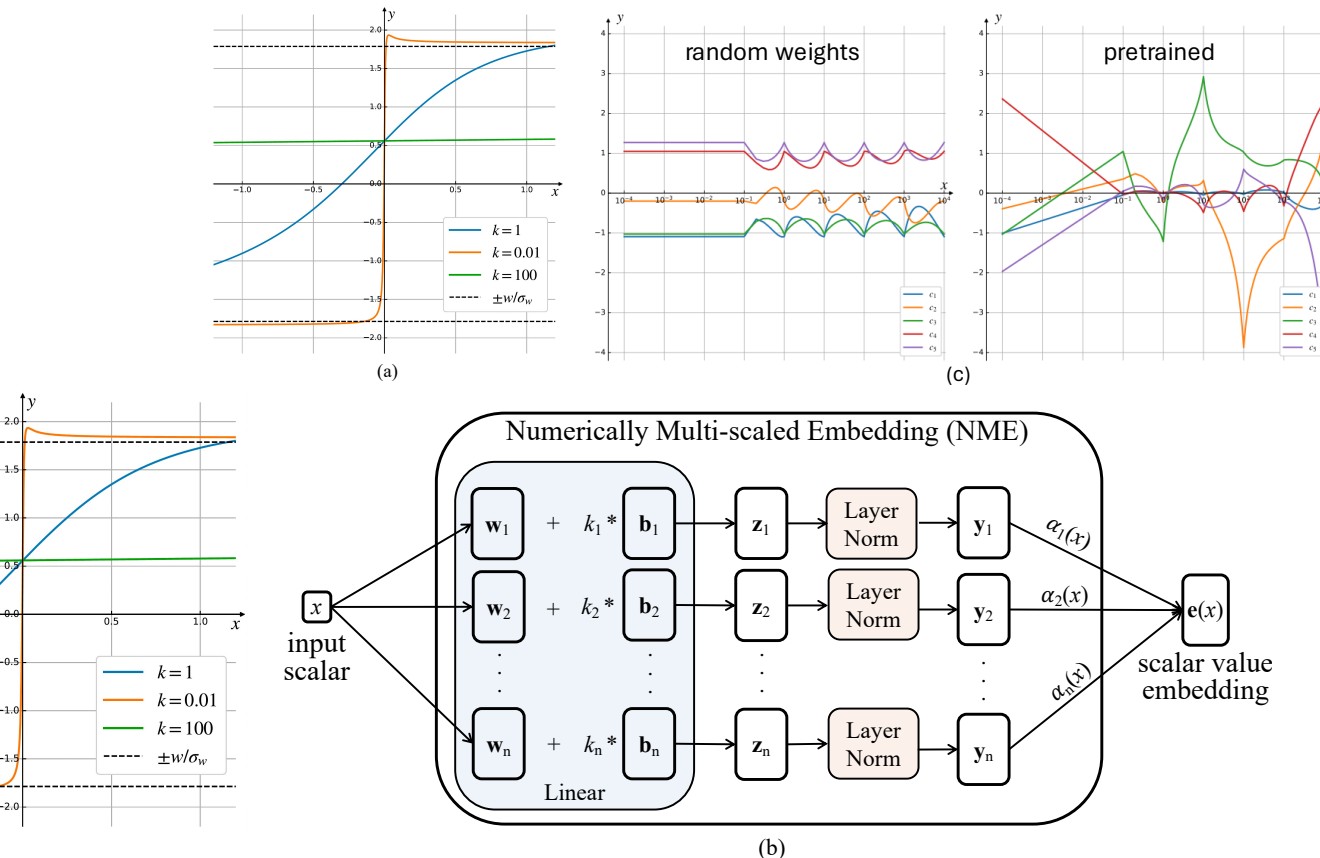

Figure 3: (a) **Output of a Basic Building Block**. The input and the output response for the basic building block of a linear layer and a LayerNorm with different multipliers $k$ set for the bias term. Only a single channel for the output is visualized. The function would saturate when the input is out of a scale related to $k$. (b) **Numerically Multi-scaled Embedding**. To encode an input scalar in arbitrary values, the proposed numerically multi-scaled embedding (NME) module ensembles outputs of multiple basic building blocks with different multipliers $k$ by a weighted average. (c) **Input and Output Response Curves of the NME Module**. The two figures show 5 channels of the output $\mathbf{e}(x)$, of a randomly initialized module and a pretrained module under log scale. The embedding module models a complex function which reflects multiple scales of variations.

and are assumed to be constant $\mathbf{1}$ and $\mathbf{0}$ here for simplicity. Note that we add a multiplier $k$ to the bias parameter. This would help us understand the behavior of the embedding module.

We first notice that the output $\mathbf{y}$ with respect to $x$ is no longer a linear function. In Figure 3(a), we plot one channel of the output $\mathbf{y}$ as a function of $x$ for different $k$ values. The parameters $\mathbf{w}$ and $\mathbf{b}$ are randomly initialized. When $|x| \gg |k|$, $\mathbf{y}$ will converge to constants $\pm\mathbf{w}/\sigma_{\mathbf{w}}$; and when $|x| \ll |k|$, $\mathbf{y}$ will converge to constants $\pm\mathbf{b}/\sigma_{\mathbf{b}}$. This means that $\mathbf{y}$ *would fail to encode anything about $x$ when $x$ is significantly larger or smaller than a scale defined by $k$.*

**Numerically Multi-scaled Embedding** Given the above analysis for the basic building block, we would need to set a proper $k$ at a scale similar to the input $x$. However, one cannot simply set $k = x$, because the overall function will cancel out $x$. We choose to enumerate all possible scales and ensemble the embeddings across scales. Let $\mathbf{y}_i(x)$ denote the embedding of input scalar $x$ at scale $k_i$. The numerically multi-scaled embedding (NME) $\mathbf{e}(x)$ is defined as

$$\mathbf{e}(x) = \sum_{i=1}^{n} \alpha_i(x) \cdot \mathbf{y}_i(x), \quad \text{where} \quad \alpha_i(x) = \frac{|\log^{-1}(|x|/k_i + \epsilon)|}{\sum_{j=1}^{n} |\log^{-1}(|x|/k_j + \epsilon)|}. \tag{3}$$

$\alpha_i$ is a weighting term based on the proportion between $x$ and $k_i$, $n$ is the number of ensembled embeddings, and $\epsilon$ is a small number to avoid computation error. Ablation on the weighted average is presented in Appendix C. We densely set the value of $k_i$ as $10^{-4}, 10^{-3}, ..., 1, ..., 10^3, 10^4$, so that the range of $k_i$ is wide enough to cover the mean and std statistics across the whole training dataset while the scale factor of 10 is determined empirically. The computation complexity of the proposed embedding module is negligible compared with the Transformer model, as it only involves an extra few paralleled linear layers.

With the proposed numerically multi-scaled embedding, we can represent arbitrary scalar values into a normalized vector space, which ensures that gradients for learning could back-propagate smoothly.

### 3.4 Extension to Multivariate Data

For multivariate time-series, we encode each window for each time-series channel independently using the aforementioned method. Parameters for encoding each window are shared across multivariate channels. Then, embeddings for each window are concatenated across channels and an additional linear layer (which is not shown in Figure 3) follows to transform them to the Transformer feature dimension. The resultant embeddings are fed to the Transformer encoder.

## 4 Experiments

### 4.1 Experimental Settings

**Pretraining Dataset**   Existing datasets for time-series analysis are relatively small individually. To address this limitation and facilitate large-scale pretraining, we propose to merge several existing datasets into a unified dataset. We consider three main sources: (1) UCR time series archive (Dau et al., 2019), (2) UEA time series archive (Bagnall et al., 2018) and (3) eight additional datasets used in recent technical papers (Eldele et al., 2021b; Zhang et al., 2022). The original training and testing splits of these datasets are retained, and only the training portions are merged. The merged dataset consists of approximately 1.89 million univariate sequences for training. With a crop size of 512 and a window size of 16, the training set amounts to about 60 million tokens. Details of the three data sources are provided below.

(1) The UCR time series archive (Dau et al., 2019) contains 128 univariate time series datasets from various sources, including sensor, motion, trajectory, etc. In total, there are 60,555 in the training set of these 128 sub-datasets by their official split.

(2) The UEA benchmark (Bagnall et al., 2018) contains 30 datasets with a wide range of cases, dimensions and series lengths for multivariate time series classification. Multivariate data in the UEA archive is partitioned into univariate sequences for joint self-supervised pretraining. This finally leads to 1,386,874 sequences for training.

(3) Other commonly used datasets in recent technical papers (Eldele et al., 2021b; Zhang et al., 2022) include: EPILEPSY (Andrzejak et al., 2001), SLEEPEEG (Kemp et al., 2000), HAR (Anguita et al., 2013), GESTURE (Liu et al., 2009), FD-A (Lessmeier et al., 2016), FD-B (Lessmeier et al., 2016), ECG (Clifford et al., 2017) and EMG (Goldberger et al., 2000). These datasets in total contain 441,757 training sequences. More information about these datasets is included in Appendix A.

**Pretraining Objective**   For self-supervised pretraining, we adopt the Bootstrap Your Own Latent (BYOL) (Grill et al., 2020) objective for its simplicity and effectiveness. In BYOL, two views of the input after data augmentation are fed to a Siamese network, where the base encoder is trained to predict the representation of the momentum encoder. The base encoder is followed by an additional predictor to avoid representation collapse. Please refer to the original paper for more details.

**Implementation Details**   We adopt a 6-layer and 8-head standard Transformer encoder with fixed sinusoidal positional encoding (Vaswani et al., 2017) as the backbone for our experiments. It uses 128-dimensional latent vectors through all of its layers, with 512 dimensions for the MLP hidden layer size. The window size for input patches is 16. For the numerically multi-scaled embedding, we choose to use 9 scales, which range from $10^{-4}$ to $10^4$ by factors of 10.

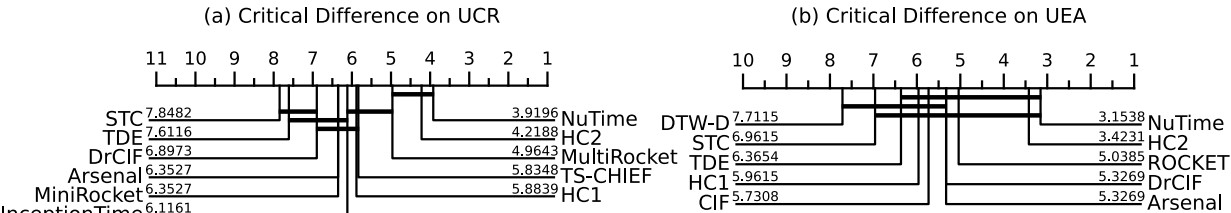

Figure 4: Test accuracy critical difference diagrams of NuTime's performance versus supervised state-of-the-art methods across (a) 112 datasets from the UCR archive and (b) 26 datasets from the UEA archive. We report the results of NuTime as an ensemble of five runs, each finetuned with different random seeds using the same pretrained model.

For pretraining, we simply choose the data augmentation of "random resized crop" for the BYOL objective. It randomly crops a sub-sequence from the original data between the range of 80% to 100%, and subsequently resizes the selected sub-sequence to a length of 512 using bilinear interpolation. The learning rate is 2e-3 for a batch size of 2048. The model is trained for a total of 100 epochs with a linear learning rate warm-up in the first 10 epochs of training and a cosine learning rate decay scheduler (Loshchilov & Hutter, 2017) with an end rate of zero. For optimization, we use AdamW (Loshchilov & Hutter, 2018) with $\beta_1 = 0.9$, $\beta_2 = 0.999$ and a weight decay of 0.05. The pretraining takes 6 hours on 4 V100 GPUs.

We transfer the pretrained model downstream classification tasks by full finetuning. The finetuning takes 100 epochs with a learning rate of 2e-4 by default. Since the model is pretrained on univariate data, an additional linear layer is added in the data embedding module for the multivariate classification tasks. For all the experiments, we report the top-1 accuracy (%, "Acc." for short) and macro F1 score (%) on the test set using the best model on the validation set.

## 4.2 Univariate Time Series Classification

**Compare with Supervised Baselines**   We first evaluate our model for univariate time series classification on 112 sub-datasets from the UCR archive. The 112 sub-datasets are chosen to exclude datasets containing series of unequal length or missing values following the practice of HIVE-COTE2.0 (HC2) (Middlehurst et al., 2021b). The state-of-the-art method HC2 is a heavily engineered system that ensembles a distinct set of classifiers: the shapelet-based classifiers (Bostrom & Bagnall, 2015), the ensemble of convolution-based classifiers (Dempster et al., 2020), the dictionary-based representation TDE (Middlehurst et al., 2021a) and the interval-based DrCIF (Middlehurst et al., 2020). Moreover, HC2 takes 1,500 epochs to train the learning-based part in its ensemble system. Due to great domain expertise being engineered into the state-of-the-art methods, only one deep learning method InceptionTime (Ismail Fawaz et al., 2020) can rank in the top 10 of the leaderboard. No prior self-supervised time series models perform close to HC2 and related methods. For fair comparisons, we ensemble 5 runs of results, each finetuned for 500 epochs with different random seeds using the same pretrained model. As shown in the critical difference diagram in Figure 4(a), NuTime achieves first place on this challenging benchmark. This is the first time that a pretrained model with a transfer learning pipeline outperforms domain-specific features and classifiers. Detailed comparisons with other methods are shown in Figure 5. Full results for these 112 datasets are in Appendix D.1.

**Compare with Self-Supervised Baselines**   To compare with previous self-supervised representation learning methods, we consider the downstream classification tasks on 125 UCR datasets, Epilepsy, FD-B and EMG following Yue et al. (2022) and Zhang et al. (2022). The baseline methods conduct unsupervised learning on individual small datasets containing only thousands of sequences, and evaluate the learned representation on the test set via finetuning. Our approach is the first one that is able to pretrain across datasets with high diversity, and is also the simplest one in terms of minimum data augmentations and a simple BYOL learning objective. The results are summarized in Table 1, and full results for the 125 UCR sub-datasets are in Appendix D.1. The reported performance for our model is the average performance of 5 independent runs. Our NuTime model outperforms the baselines on all the tested benchmarks.

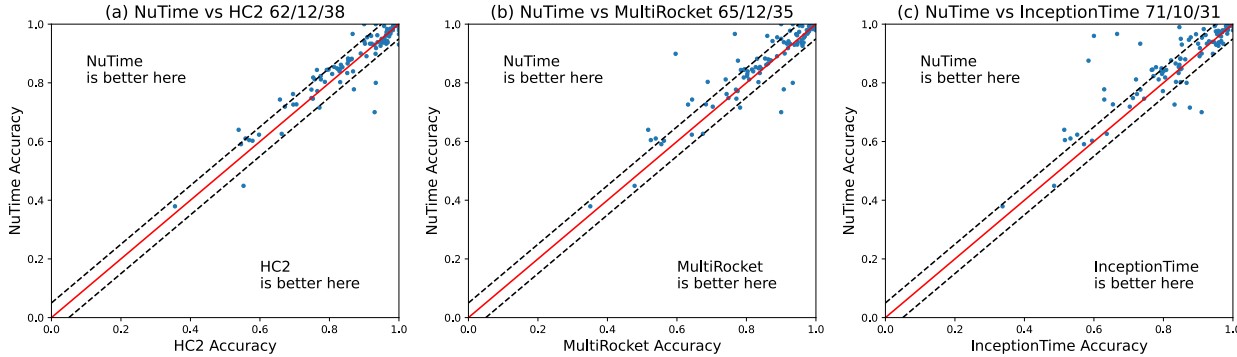

Figure 5: Accuracy comparison of NuTime and (a) HIVE-COTE2.0 (Middlehurst et al., 2021b), (b) MultiRocket (Tan et al., 2022), (c) InceptionTime (Ismail Fawaz et al., 2020) on 112 datasets from the UCR archive. Each subfigure's title displays a win/tie/loss comparison between NuTime and other methods. The two dotted lines indicate the 5% interval.

Table 1: Performance comparisons with self-supervised models for univariate time series classification.

| Method | 125 UCR Datasets | EPILEPSY | | FD-B | | EMG | |
|---|---|---|---|---|---|---|---|
| | Avg. Acc. | Acc. | Macro-F1 | Acc. | Macro-F1 | Acc. | Macro-F1 |
| TNC | 76.1 | - | - | - | - | - | - |
| T-Loss | 80.6 | - | - | - | - | - | - |
| TS-TCC | 75.7 | $92.53_{\pm0.98}$ | $86.33_{\pm2.15}$ | $54.99_{\pm2.20}$ | $54.18_{\pm3.38}$ | $78.89_{\pm1.92}$ | $59.04_{\pm9.52}$ |
| TS2Vec | 83.0 | $93.95_{\pm0.44}$ | $90.45_{\pm0.67}$ | $47.90_{\pm1.13}$ | $43.89_{\pm1.07}$ | $78.54_{\pm3.18}$ | $67.66_{\pm5.01}$ |
| TF-C | - | $94.95_{\pm1.08}$ | $91.49_{\pm5.34}$ | $69.38_{\pm2.31}$ | $74.87_{\pm2.68}$ | $81.71_{\pm2.87}$ | $76.83_{\pm3.11}$ |
| Ti-MAE | 82.32 | 89.71 | 68.55 | 70.88 | 66.56 | 69.99 | 70.89 |
| SimMTM | - | 95.49 | 92.81 | 69.40 | 75.11 | 97.56 | 98.14 |
| Ours | $\mathbf{86.9}_{\pm0.1}$ | $\mathbf{95.73}_{\pm0.10}$ | $\mathbf{93.11}_{\pm0.16}$ | $\mathbf{92.86}_{\pm2.04}$ | $\mathbf{93.64}_{\pm1.99}$ | $\mathbf{100.0}_{\pm0.0}$ | $\mathbf{100.0}_{\pm0.0}$ |

### 4.3   Multivariate Time Series Classification

**Compare with Supervised Baselines**   We transfer the same pretrained model on univariate time-series data to multivariate classification benchmarks by an extension to the data embedding module described in Section 3.4. We evaluate its performance on the UEA archive and compare it with state-of-the-art techniques, which are domain-specified supervised methods. The critical difference diagram and detailed comparisons to previous methods are shown in Figure 4(b) and Figure 6. Detailed results on 26 datasets are in Appendix D.2. NuTime achieves first place on this challenging benchmark against heavily engineered competitors, demonstrating that the pretrained model successfully learns a transferable representation from single-dimensional data to multi-dimensional data.

**Compare with Self-supervised Baselines**   We also compare NuTime with strong self-supervised representation learning models. The results are summarized in Table 2, and full results are shown in Appendix D.2. Our model outperforms others by scaling the pretraining data effectively, even with a simple data augmentation and simple contrastive learning objective.

### 4.4   Few-Shot Learning

One critical capability for a large-scale representation learning model is the few-shot generalization. We follow Narwariya et al. (2020) for a few-shot time-series benchmark using 41 datasets from the UCR archive. We consider the 5-shot learning scenario and 100 episodes are drawn from each dataset. By finetuning the pretrained model with few-shot data, NuTime outperforms baselines including 1NN based on Euclidean distance and dynamic time warping (DTW), BOSS (Bag-of-SFA-Symbols) (Schäfer, 2015), ResNet (He et al.,

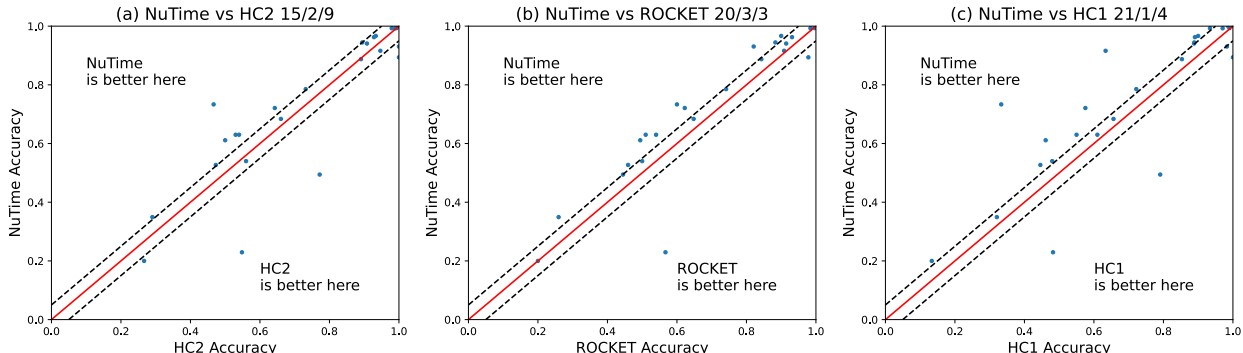

Figure 6: Accuracy comparison of NuTime and (a) HIVE-COTE2.0 (Middlehurst et al., 2021b), (b) ROCKET (Dempster et al., 2020) and (c) HIVE-COTE1.0 (Lines et al., 2016) on 26 datasets from the UEA archive. Each subfigure's title displays a win/tie/loss comparison between NuTime and other methods. The two dotted lines indicate the 5% interval.

Table 2: Performance comparisons with self-supervised models for multivariate time series classification.

| Method | 29 UEA Datasets | | GESTURE | |
|---|---|---|---|---|
| | Avg. Acc. | Avg. Rank | Acc. | Macro-F1 |
| TS-TCC | 68.2 | 4.5 | $71.88_{\pm 3.49}$ | $69.84_{\pm 3.60}$ |
| TS2Vec | 71.2 | 3.2 | $69.17_{\pm 3.33}$ | $65.70_{\pm 3.92}$ |
| TF-C | - | - | $76.42_{\pm 1.96}$ | $75.72_{\pm 3.11}$ |
| Ti-MAE | - | - | 71.88 | 68.37 |
| SimMTM | - | - | **80.00** | 78.67 |
| Ours | $\mathbf{77.8}_{\pm 0.4}$ | **1.5** | $\mathbf{80.00}_{\pm 1.36}$ | $\mathbf{78.97}_{\pm 0.90}$ |

Table 3: Few-shot learning results (5 shots) on the UCR archive.

| Method | Acc. |
|---|---|
| 1NN | 56.6 |
| DTW | 61.8 |
| BOSS | 62.5 |
| ResNet-Scratch | 62.7 |
| FS-1 | 65.3 |
| FS-2 | 66.3 |
| Ours | **67.5** |

2016) trained from scratch and dedicate meta-learning methods designed for few-shot adaption, FS-1 and FS-2 (Narwariya et al., 2020). The results are summarized in Table 3 with details shown in Appendix D.3.

### 4.5  Additional Downsteam Tasks

Besides univariate and multivariate classification, we also evaluate the proposed NuTime model on additional downstream tasks including clustering and anomaly detection. Experimental results are provided in Appendix B. NuTime consistently outperforms all other methods by a significant margin in both these downstream tasks. This suggests that our proposed method is capable of learning distinctive and generalizable time-series representations from a large-scale dataset, even with a simple and straightforward self-supervised approach.

### 4.6  Ablation Study

Ablation studies are conducted on 128 UCR datasets. We report the average performance by either finetuning the model from the pretrained checkpoint or training the model from scratch. All results are the average of 5 runs.

**Data Normalization and Encoding**  We first study various ways to preprocess and normalize the data by keeping the overall Transformer backbone. We consider Z-score, instance normalization (IN) and no preprocessing at all (i.e., identity) for the input sequence. A linear layer and a LayerNorm are used to encode windows to tokens. We also consider alternative methods proposed in PLE (Gorishniy et al., 2022) to encode the mean and the std scalars. In Table 4, our numerically multi-scaled embedding outperforms all the baselines. The PLE method relies on the quantiles of the training dataset and thus is difficult to scale properly when the data is complex and large.

Table 4: Ablation study for various data encoding and normalization methods.

| Encoding | Fine-tune | | From Scratch | |
|---|---|---|---|---|
| | Acc. | Macro-F1 | Acc. | Macro-F1 |
| Z-score | $80.88_{\pm 0.13}$ | $76.75_{\pm 0.46}$ | $76.59_{\pm 0.55}$ | $71.26_{\pm 0.92}$ |
| IN | $79.97_{\pm 0.54}$ | $75.47_{\pm 0.84}$ | $75.38_{\pm 0.60}$ | $69.73_{\pm 0.98}$ |
| Identity | $82.02_{\pm 0.29}$ | $78.12_{\pm 0.40}$ | $73.53_{\pm 0.30}$ | $67.13_{\pm 0.53}$ |
| PLE-Q | $84.41_{\pm 0.10}$ | $81.97_{\pm 0.18}$ | $77.86_{\pm 0.59}$ | $72.57_{\pm 0.80}$ |
| PLE-T | $83.59_{\pm 0.14}$ | $80.97_{\pm 0.19}$ | $73.25_{\pm 0.69}$ | $66.84_{\pm 1.05}$ |
| PLE-P | $84.20_{\pm 0.29}$ | $81.69_{\pm 0.37}$ | $69.34_{\pm 0.70}$ | $62.43_{\pm 0.70}$ |
| Ours | $\mathbf{86.87}_{\pm 0.11}$ | $\mathbf{84.74}_{\pm 0.18}$ | $\mathbf{79.30}_{\pm 0.84}$ | $\mathbf{74.09}_{\pm 1.48}$ |

Table 5: Ablation study for varying the number of scales in the numerical embedding module.

| Num. Scales | Fine-tuned | | From Scratch | |
|---|---|---|---|---|
| | Acc. | Macro-F1 | Acc. | Macro-F1 |
| 0 | $80.48_{\pm 0.05}$ | $77.43_{\pm 0.10}$ | $68.14_{\pm 0.40}$ | $61.43_{\pm 0.62}$ |
| 1 | $86.45_{\pm 0.20}$ | $84.01_{\pm 0.36}$ | $74.19_{\pm 0.54}$ | $67.64_{\pm 1.23}$ |
| 3 | $86.64_{\pm 0.18}$ | $84.11_{\pm 0.29}$ | $78.32_{\pm 0.66}$ | $73.01_{\pm 0.92}$ |
| 5 | $86.74_{\pm 0.13}$ | $84.35_{\pm 0.33}$ | $\mathbf{79.30}_{\pm 0.84}$ | $\mathbf{74.09}_{\pm 1.48}$ |
| 7 | $86.79_{\pm 0.114}$ | $84.57_{\pm 0.134}$ | $78.89_{\pm 0.70}$ | $73.72_{\pm 0.78}$ |
| 9 | $\mathbf{86.87}_{\pm 0.11}$ | $\mathbf{84.74}_{\pm 0.18}$ | $78.58_{\pm 0.45}$ | $73.13_{\pm 0.95}$ |

Table 6: Ablation study on window size and mean/std embedding dimensions. The default window size and mean/std embedding dimension in the main paper are set to 16 and 32 respectively.

| Window size | 4 | 8 | 16 | 32 | 64 | 128 |
|---|---|---|---|---|---|---|
| Avg. Accuracy | $84.66_{\pm 0.31}$ | $85.24_{\pm 0.32}$ | $\mathbf{86.87}_{\pm 0.11}$ | $85.12_{\pm 0.45}$ | $84.22_{\pm 0.14}$ | $82.69_{\pm 0.03}$ |
| Avg. Macro-F1 | $81.60_{\pm 0.50}$ | $82.43_{\pm 0.34}$ | $\mathbf{84.74}_{\pm 0.18}$ | $82.58_{\pm 0.46}$ | $81.59_{\pm 0.25}$ | $79.93_{\pm 0.16}$ |

| Emb. dimension | 4 | 8 | 16 | 32 | 64 | 128 |
|---|---|---|---|---|---|---|
| Avg. Accuracy | $85.96_{\pm 0.16}$ | $86.19_{\pm 0.07}$ | $86.08_{\pm 0.09}$ | $\mathbf{86.87}_{\pm 0.11}$ | $85.91_{\pm 0.20}$ | $86.00_{\pm 0.15}$ |
| Avg. Macro-F1 | $83.09_{\pm 0.25}$ | $83.51_{\pm 0.27}$ | $83.22_{\pm 0.21}$ | $\mathbf{84.74}_{\pm 0.18}$ | $83.00_{\pm 0.17}$ | $83.10_{\pm 0.39}$ |

Additionally, we also consider an improved instance normalization baseline where extra tokens encoding the instance mean and std are appended to the Transformer network. Such approach improves the fine-tune results from 79.97 to 83.3 by accuracy and from 75.47 to 79.65 by Macro-F1 score. However, it still falls behind our approach by a wide margin. This demonstrates the benefits of window-wise normalization over instance-wise normalization.

**Number of Numerical Scales** We vary the number of multipliers $k$ in our multi-scaled numerical data embedding module. In Table 5, the performance improves from a single scale to 9 scales for the transfer setting. This shows that the capability to encode multi-scaled data is critical and our method provides an effective solution. Training from scratch attains its best results with 5 scales. Since the individual datasets are relatively small, this number of scales is sufficient in this case.

**Window size and Embedding Dimensions** We provide the ablation study on window size and mean/std embedding dimension in Table 6. The default window size and mean/std embedding dimension used in the main paper are set to 16 and 32 respectively.

**Pretraining Data** As many works study self-supervised transfer learning within the same domain (Eldele et al., 2021b; Yue et al., 2022; Zhang et al., 2022), we also pretrain our model on each individual dataset from the UCR archive and evaluate the performance on the same dataset. It achieves an accuracy of 79.7% and a Macro-F1 score of 74.8%, which is much lower than our large-scale pretraining results of **86.9**% accuracy and **84.7**% Macro-F1 score, which suggests that our model successfully learns a transferable representation from large-scale data.

## 5 Conclusion

In this paper, we propose the NuTime model for large-scale time series pretraining. The model is based on the Transformer architecture, which takes input as a set of tokens from non-overlapping windows. Each window is represented by its normalized shape, the window mean and the window standard deviation. We develop a multi-scaled numerical embedding method for representing the scalar values of mean and std. The model can take raw values of time-series data as input without any data normalization and transformation. To demonstrate that the proposed model can learn numerical structures with different scales of variations, we conduct the first large-scale pretraining on a dataset with great domain diversity. The pretrained model achieves state-of-the-art performance when transferred to downstream classification benchmarks. We hope that this work will pave the way to general-purpose foundation models for time-series analysis.

**Limitations and Discussions** The learned representation of the model may be subjective to the biases and inequalities from the training data. Also, the model might introduce unexpected behaviors on data it never sees during training. The optimal hyper-parameters of the model may vary according to the training dataset. Thus, training the NuTime model on other dataset may require additional hyper-parameter tuning, such as window size, and pre-defined numerical scales.

The proposed method aims to effectively encode time-series data from diverse domains. It is not yet able to decode the representation to a numerical value at the original scale, and thus it is not directly applicable for forecasting problems. Straight-forward decoding through linear layers assumes a single numerical scale of the output space. In order to decode the vector representation to numerical values of arbitrary scales, we hypothesize that an iterative procedure needs to be developed. For example, diffusion models decodes RGB values in an iterative fashion and RAFT (Teed & Deng, 2020) predicts optical flow values with an recurrent network. However, decoding time series values may encounter even bigger challenges as the numerical variations are a lot higher than RGB pixels and optical flows. We wish to tackle this problem in the future.

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

# A   More Information about Datasets

(1) EPILEPSY (Andrzejak et al., 2001) is an epilepsy seizure recognition dataset composed of EEG recordings for 500 subjects labeled by 5 categories with 9,200 training and 2,300 test samples. 4 classes that do not have an epileptic seizure in this dataset are merged into one class. Thus, each time series sample has a binary label. To compare the proposed NuTime model with other self-supervised methods, we use the dataset split by Zhang et al. (2022), having 60 samples for training, 20 samples for validation, and 11,420 samples for testing.

(2) SLEEPEEG (Kemp et al., 2000) contains 371,055 training and 90,315 test whole-night sleeping electroencephalography (EEG) recordings that are classified into 5 patterns including Wake (W), Non-rapid eye movement (N1, N2, N3) and Rapid Eye Movement (REM).

(3) HAR (Anguita et al., 2013) is a multivariate time series with 9 channels that are sensor readings for 30 subjectives when they are performing 6 actions like walking, sitting and lying down. There are 5,881 and 2,947 samples in the training and test dataset respectively.

(4) GESTURE (Liu et al., 2009) is a set of eight simple gestures generated from accelerators consisting of the XYZ coordinates of each motion, and it contains 320 and 120 training and test samples respectively.

(5) FD-A and (6) FD-B (Lessmeier et al., 2016) is a real-world fault diagnosis dataset generated by an electromechanical drive system with 2 fault classes and 1 health class in different conditions of rolling bearings. FD-A and (6) FD-B refer to the data collected under condition A and condition B. For a fair comparison, we adopt the same training and test split as Zhang et al. (2022), and there are 60 and 13,559 samples in FD-B training and test dataset for classification benchmarking.

(7) ECG (Clifford et al., 2017) contains 8,528 electrocardiogram (ECG) recordings into four categories that indicate different underlying conditions of cardiac arrhythmias. These long recordings are divided into short sequences of 5 seconds that are physiologically meaningful, and each short sequence has 1500 observations and is regarded as a single sample.

(8) EMG (Goldberger et al., 2000) is a set of electromyography (EMG) recordings from the tibialis anterior muscle from 3 subjectives, and each of them indicates a classification category. EMG is univariate and split into training and test set following Zhang et al. (2022), leading to 122 training and 41 test samples.

# B   Additional Downstream Tasks

**Clustering**   For the clustering task on a two-class dataset EPILEPSY, we employ K-means (K=2) on the learned representations from NuTime and its self-supervised counterparts. These representations are obtained after independent fine-tuning on EPILEPSY, following the settings of Zhang et al. (2022). We evaluate the clustering performance using three widely adopted metrics: Silhouette score, Adjusted Rand Index (ARI), and Normalized Mutual Information (NMI). In order to highlight the generalization of the time-series representations learned by NuTime, we also perform clustering using the NuTime representations without any fine-tuning on the downstream task data.

**Anomaly Detection**   Following the approach presented by Zhang et al. (2022), we conduct anomaly detection to identify abnormal time series samples instead of outlier observations. Specifically, we adopt a highly imbalanced subset of FD-B with 1000 samples. Among these samples, 900 correspond to undamaged bearings, while the remaining 100 samples pertain to bearings with either inner or outer damage. A one-class SVM is applied to the learned representations from NuTime and its self-supervised counterparts to perform anomaly detection.

The results for clustering and anomaly detection are presented in Table 7 and Table 8 respectively. All experiments are repeated 5 times, and we report the mean and standard deviation of all metrics across these 5 runs. Our NuTime model consistently outperforms all other models by a significant margin in both additional tasks. Remarkably, even without finetuning on the specific data of the clustering task and directly utilizing representations from the pretrained model, our NuTime model demonstrates superior performance compared to other self-supervised methods after finetuning across most metrics. This suggests

that our proposed method is capable of learning distinctive and generalizable time-series representations from a large-scale dataset, even with a simple and straightforward self-supervised approach.

To visually illustrate the distinct representations learned by NuTime (i.e., the token at position 0 of the Transformer output), we employ t-SNE (van der Maaten & Hinton, 2008) on the two datasets used for additional downstream tasks. The resulting visualizations are depicted in Figure 7.

Table 7: Performance on EPILEPSY for clustering task. All the methods are pretrained in a self-supervised manner. "Ours (w/o ft.)" denotes that the pretrained NuTime model representations are directly employed for the clustering task without undergoing finetuning, whereas the remaining methods are finetuned on the EPILEPSY dataset.

| Method | Silhouette Score | ARI | NMI |
|---|---|---|---|
| KNN | $0.1208_{\pm0.0271}$ | $0.0549_{\pm0.0059}$ | $0.0096_{\pm0.0014}$ |
| TNC | $0.2353_{\pm0.0018}$ | $0.0211_{\pm0.0061}$ | $0.0082_{\pm0.0018}$ |
| CPC | $0.2223_{\pm0.0011}$ | $0.0153_{\pm0.0196}$ | $0.0063_{\pm0.0055}$ |
| TS-TCC | $0.5154_{\pm0.0458}$ | $0.6307_{\pm0.0325}$ | $0.5178_{\pm0.0283}$ |
| TF-C | $\underline{0.5439}_{\pm0.0417}$ | $0.6583_{\pm0.0259}$ | $0.5567_{\pm0.0172}$ |
| Ours (w/o ft.) | $0.4923_{\pm0.0010}$ | $\underline{0.6854}_{\pm0.0000}$ | $\underline{0.5737}_{\pm0.0000}$ |
| Ours | $\mathbf{0.7945}_{\pm0.0016}$ | $\mathbf{0.7944}_{\pm0.0046}$ | $\mathbf{0.6450}_{\pm0.0063}$ |

Table 8: Performance on an FD-B subset for sample-level anomaly detection task. All the methods are pretrained in a self-supervised manner and finetuned on the FD-B anomaly detection subset.

| Method | Precision | Recall | F1 Score | AUROC |
|---|---|---|---|---|
| KNN | $0.4785_{\pm0.0356}$ | $0.6159_{\pm0.0585}$ | $0.5061_{\pm0.0278}$ | $0.7653_{\pm0.0153}$ |
| TNC | $0.8354_{\pm0.0484}$ | $0.7882_{\pm0.0157}$ | $0.7957_{\pm0.0165}$ | $0.8231_{\pm0.0379}$ |
| CPC | $0.5967_{\pm0.0776}$ | $0.4896_{\pm0.0866}$ | $0.5238_{\pm0.0792}$ | $0.7859_{\pm0.0356}$ |
| TS-TCC | $0.6219_{\pm0.0183}$ | $0.4431_{\pm0.0658}$ | $0.4759_{\pm0.0562}$ | $0.7966_{\pm0.0441}$ |
| TF-C | $0.8526_{\pm0.0367}$ | $0.7823_{\pm0.0299}$ | $0.8312_{\pm0.0186}$ | $0.8598_{\pm0.0283}$ |
| Ours | $\mathbf{1.0000}_{\pm0.0000}$ | $\mathbf{0.8962}_{\pm0.0376}$ | $\mathbf{0.9448}_{\pm0.0204}$ | $\mathbf{0.9481}_{\pm0.0188}$ |

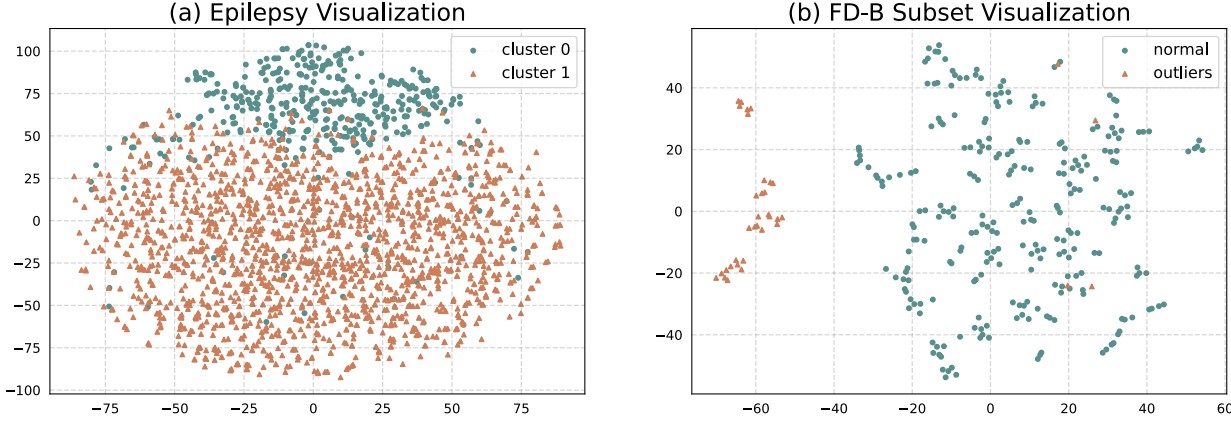

Figure 7: Visualization of the learned representations from NuTime on EPILEPSY and FD-B subset. It highlights the distinct separation of these representations, which is highly advantageous for downstream tasks, such as clustering and anomaly detection.

## C   Weighted Average in Ensemble Module

We design a heuristic weighting term within the ensemble module, which is based on the proportion between the input scalar $x$ and the provided bias multipliers $k$. It is motivated by an insight that the input scalar $x$ should be embedded by the basic building block (i.e., linear + LayerNorm) which has a $k$ that shares a comparable numerical scale with $x$. Otherwise, as the number of ensemble scales increases, the information encapsulated within $x$ would be diluted by a simple average ensemble. Figure 8 confirms this hypothesis.

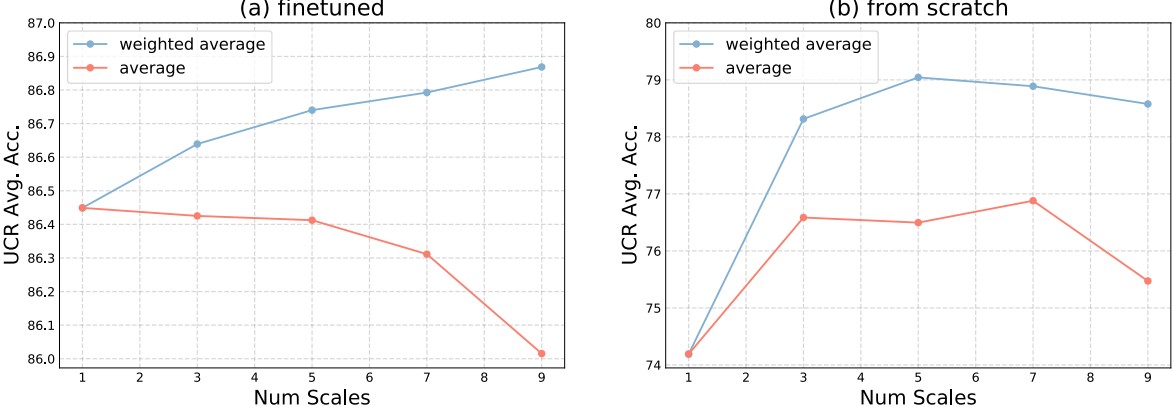

Figure 8: Average accuracy (%) on 128 UCR datasets by (a) finetuning from the pretrained checkpoint and (b) training from scratch with or without the weighting term in the ensemble module. "Num Scales" stands for the number of ensembled numerical scales.

## D   Full Results on UCR/UEA Datasets

### D.1   Univariate Time Series Classification

The full results, in terms of top-1 accuracy, for both supervised and self-supervised pretrained methods on each sub-dataset in the UCR benchmark are presented in Table 9 and Table 10 respectively.

Table 9: Full results for 112 UCR datasets of NuTime and supervised counterparts.

| Dataset | NuTime | HC2 | TS-CHIEF | HC1 | InceptionTime | MiniRocket | MultiRocket |
|---|---|---|---|---|---|---|---|
| ACSF1 | 0.700 | 0.930 | 0.840 | **0.940** | 0.910 | 0.900 | 0.900 |
| Adiac | **0.854** | 0.813 | 0.798 | 0.811 | 0.844 | 0.770 | 0.839 |
| ArrowHead | **0.880** | 0.863 | 0.806 | 0.823 | 0.863 | 0.794 | 0.863 |
| BME | **1.000** | **1.000** | 0.993 | 0.953 | **1.000** | **1.000** | **1.000** |
| Beef | **0.967** | 0.867 | 0.767 | 0.900 | 0.667 | 0.833 | 0.767 |
| BeetleFly | 0.950 | 0.950 | **1.000** | 0.950 | 0.850 | 0.900 | 0.900 |
| BirdChicken | **1.000** | 0.900 | 0.950 | 0.950 | 0.950 | 0.900 | 0.900 |
| CBF | 0.998 | **1.000** | 0.998 | 0.999 | 0.999 | **1.000** | 0.996 |
| Car | **0.933** | 0.917 | 0.867 | 0.833 | 0.900 | 0.850 | 0.917 |
| Chinatown | 0.980 | 0.983 | 0.968 | 0.980 | **0.985** | 0.980 | 0.974 |
| ChlorineConcentration | 0.716 | 0.771 | 0.660 | 0.739 | **0.876** | 0.807 | 0.782 |
| CinCECGTorso | 0.930 | **1.000** | 0.981 | 0.995 | 0.853 | 0.837 | 0.946 |
| Coffee | **1.000** | **1.000** | **1.000** | **1.000** | **1.000** | **1.000** | **1.000** |
| Computers | **0.844** | 0.764 | 0.700 | 0.776 | 0.808 | 0.752 | 0.784 |
| CricketX | 0.846 | 0.826 | 0.821 | 0.808 | **0.849** | 0.826 | 0.795 |
| CricketY | 0.833 | 0.851 | 0.800 | 0.823 | 0.844 | **0.856** | 0.838 |
| CricketZ | **0.882** | 0.864 | 0.831 | 0.823 | 0.849 | 0.859 | 0.831 |
| Crop | 0.772 | 0.765 | 0.764 | 0.773 | **0.799** | 0.751 | 0.775 |
| DiatomSizeReduction | **0.993** | 0.958 | 0.964 | 0.935 | 0.948 | 0.967 | 0.961 |
| DistalPhalanxOutlineAgeGroup | **0.777** | 0.748 | 0.748 | 0.755 | 0.734 | 0.748 | 0.770 |
| DistalPhalanxOutlineCorrect | **0.812** | 0.775 | 0.757 | 0.772 | 0.786 | 0.761 | 0.808 |
| DistalPhalanxTW | **0.727** | 0.705 | 0.676 | 0.676 | 0.655 | 0.719 | 0.683 |
| ECG200 | 0.900 | 0.890 | 0.840 | 0.860 | **0.910** | **0.910** | **0.910** |

| | | | | | | |
|---|---|---|---|---|---|---|
| ECG5000 | 0.940 | 0.947 | 0.946 | **0.947** | 0.942 | **0.947** | 0.947 |
| ECGFiveDays | 0.979 | **1.000** | **1.000** | **1.000** | **1.000** | **1.000** | **1.000** |
| EOGHorizontalSignal | **0.743** | 0.657 | 0.655 | 0.610 | 0.630 | 0.652 | 0.644 |
| EOGVerticalSignal | **0.605** | 0.569 | 0.597 | 0.547 | 0.517 | 0.533 | 0.525 |
| Earthquakes | 0.748 | **0.748** | **0.748** | **0.748** | 0.712 | **0.748** | **0.748** |
| ElectricDevices | **0.812** | 0.758 | 0.760 | 0.746 | 0.721 | 0.726 | 0.740 |
| EthanolLevel | 0.726 | 0.696 | 0.528 | 0.684 | **0.832** | 0.570 | 0.632 |
| FaceAll | 0.839 | 0.868 | 0.842 | 0.779 | 0.793 | **0.947** | 0.804 |
| FaceFour | 0.966 | **1.000** | **1.000** | 0.989 | 0.955 | 0.977 | 0.943 |
| FacesUCR | 0.933 | 0.963 | 0.967 | 0.959 | **0.972** | 0.961 | 0.961 |
| FiftyWords | **0.862** | 0.833 | 0.848 | 0.776 | 0.846 | 0.833 | 0.857 |
| Fish | **0.994** | 0.989 | 0.994 | 0.994 | 0.983 | 0.971 | 0.983 |
| FordA | 0.937 | 0.954 | 0.950 | 0.951 | **0.961** | 0.942 | 0.958 |
| FordB | 0.802 | 0.831 | 0.823 | 0.832 | **0.849** | 0.806 | 0.835 |
| FreezerRegularTrain | 0.999 | **1.000** | 0.998 | 0.998 | 0.997 | 0.996 | **1.000** |
| FreezerSmallTrain | 0.990 | **0.999** | 0.998 | 0.986 | 0.846 | 0.950 | 0.995 |
| GunPoint | **1.000** | **1.000** | **1.000** | **1.000** | **1.000** | 0.993 | **1.000** |
| GunPointAgeSpan | **1.000** | 0.997 | **1.000** | 0.997 | 0.991 | 0.997 | **1.000** |
| GunPointMaleVersusFemale | **1.000** | **1.000** | **1.000** | **1.000** | **1.000** | 0.997 | **1.000** |
| GunPointOldVersusYoung | **1.000** | **1.000** | **1.000** | **1.000** | **1.000** | 0.994 | **1.000** |
| Ham | **0.762** | 0.695 | 0.705 | 0.705 | 0.724 | 0.714 | 0.743 |
| HandOutlines | **0.962** | 0.938 | 0.938 | 0.932 | 0.959 | 0.943 | 0.946 |
| Haptics | **0.591** | 0.545 | 0.529 | 0.539 | 0.571 | 0.519 | 0.555 |
| Herring | **0.719** | 0.672 | 0.641 | 0.672 | 0.703 | 0.672 | 0.703 |
| HouseTwenty | 0.983 | 0.966 | 0.975 | 0.983 | 0.916 | 0.966 | **0.992** |
| InlineSkate | 0.449 | **0.553** | 0.527 | 0.485 | 0.485 | 0.475 | 0.478 |
| InsectEPGRegularTrain | **1.000** | **1.000** | **1.000** | **1.000** | **1.000** | **1.000** | **1.000** |
| InsectEPGSmallTrain | **1.000** | **1.000** | **1.000** | **1.000** | **1.000** | 0.984 | **1.000** |
| InsectWingbeatSound | 0.626 | 0.663 | 0.644 | 0.660 | 0.637 | 0.661 | **0.675** |
| ItalyPowerDemand | 0.958 | **0.972** | 0.965 | 0.960 | 0.966 | 0.969 | 0.968 |
| LargeKitchenAppliances | 0.891 | **0.907** | 0.768 | 0.853 | 0.904 | 0.904 | 0.888 |
| Lightning2 | 0.803 | 0.787 | **0.836** | 0.770 | **0.836** | 0.754 | 0.689 |
| Lightning7 | **0.849** | 0.822 | 0.767 | 0.726 | 0.795 | 0.822 | 0.836 |
| Mallat | 0.977 | **0.978** | 0.971 | 0.973 | 0.958 | 0.957 | 0.925 |
| Meat | **1.000** | 0.933 | 0.883 | 0.917 | 0.950 | 0.950 | 0.933 |
| MedicalImages | **0.830** | 0.808 | 0.796 | 0.732 | 0.796 | 0.793 | 0.803 |
| MiddlePhalanxOutlineAgeGroup | 0.623 | 0.597 | 0.571 | 0.591 | 0.552 | 0.591 | **0.643** |
| MiddlePhalanxOutlineCorrect | **0.866** | 0.852 | 0.825 | 0.835 | 0.835 | 0.838 | 0.863 |
| MiddlePhalanxTW | **0.610** | 0.558 | 0.565 | 0.584 | 0.532 | 0.565 | 0.539 |
| MixedShapesRegularTrain | 0.979 | 0.976 | 0.971 | 0.969 | 0.970 | 0.967 | **0.981** |
| MixedShapesSmallTrain | 0.954 | 0.959 | 0.949 | 0.954 | 0.911 | 0.938 | **0.965** |
| MoteStrain | 0.966 | **0.970** | 0.927 | 0.968 | 0.894 | 0.915 | 0.945 |
| NonInvasiveFetalECGThorax1 | 0.926 | 0.947 | 0.917 | 0.924 | 0.958 | 0.955 | **0.961** |
| NonInvasiveFetalECGThorax2 | 0.937 | 0.967 | 0.948 | 0.945 | 0.959 | **0.969** | 0.964 |
| OSULeaf | 0.963 | 0.963 | **0.996** | 0.983 | 0.921 | 0.942 | 0.967 |
| OliveOil | 0.800 | **0.933** | 0.900 | 0.867 | 0.867 | **0.933** | **0.933** |
| PhalangesOutlinesCorrect | 0.848 | 0.831 | 0.818 | 0.821 | **0.852** | 0.834 | 0.843 |
| Phoneme | 0.379 | 0.355 | 0.361 | **0.381** | 0.337 | 0.283 | 0.351 |
| PigAirwayPressure | 0.899 | 0.933 | **0.976** | 0.962 | 0.942 | 0.091 | 0.596 |
| PigArtPressure | 0.995 | 0.995 | 0.981 | 0.990 | **1.000** | 0.952 | 0.952 |
| PigCVP | 0.942 | **0.966** | 0.962 | 0.962 | **0.966** | 0.938 | 0.875 |
| Plane | **1.000** | **1.000** | **1.000** | **1.000** | **1.000** | **1.000** | **1.000** |
| PowerCons | 0.994 | 0.983 | 0.989 | **1.000** | 0.994 | 0.933 | 0.983 |
| ProximalPhalanxOutlineAgeGroup | **0.883** | 0.854 | 0.859 | 0.839 | 0.849 | 0.859 | 0.854 |
| ProximalPhalanxOutlineCorrect | **0.942** | 0.900 | 0.883 | 0.890 | 0.931 | 0.900 | 0.921 |
| ProximalPhalanxTW | **0.839** | 0.829 | 0.824 | 0.824 | 0.800 | 0.820 | 0.820 |
| RefrigerationDevices | **0.640** | 0.539 | 0.571 | 0.579 | 0.515 | 0.536 | 0.517 |
| Rock | **0.960** | 0.920 | 0.900 | **0.960** | 0.600 | 0.900 | 0.860 |
| ScreenType | **0.603** | 0.579 | 0.499 | 0.557 | 0.595 | 0.477 | 0.563 |
| SemgHandGenderCh2 | 0.947 | 0.957 | 0.923 | **0.970** | 0.873 | 0.918 | 0.958 |
| SemgHandMovementCh2 | 0.876 | 0.856 | **0.878** | 0.871 | 0.584 | 0.638 | 0.776 |
| SemgHandSubjectCh2 | 0.933 | 0.902 | 0.924 | **0.953** | 0.733 | 0.889 | 0.924 |
| ShapeletSim | **1.000** | **1.000** | **1.000** | **1.000** | 0.994 | **1.000** | **1.000** |
| ShapesAll | **0.933** | 0.922 | 0.925 | 0.932 | 0.925 | 0.907 | 0.925 |
| SmallKitchenAppliances | **0.853** | 0.837 | 0.824 | 0.835 | 0.779 | 0.813 | 0.827 |
| SmoothSubspace | 0.993 | 0.987 | **1.000** | 0.980 | 0.980 | 0.980 | 0.980 |
| SonyAIBORobotSurface1 | **0.955** | 0.908 | 0.832 | 0.767 | 0.879 | 0.923 | 0.889 |
| SonyAIBORobotSurface2 | 0.930 | 0.930 | 0.896 | 0.942 | **0.952** | 0.915 | 0.938 |

| | | | | | | | |
|---|---|---|---|---|---|---|---|
| StarLightCurves | 0.981 | 0.982 | **0.983** | 0.981 | 0.978 | 0.981 | 0.982 |
| Strawberry | 0.968 | 0.976 | 0.970 | 0.970 | **0.984** | 0.981 | 0.976 |
| SwedishLeaf | 0.973 | 0.965 | 0.965 | 0.954 | 0.970 | 0.963 | **0.978** |
| Symbols | 0.985 | 0.976 | 0.978 | 0.980 | **0.985** | 0.973 | 0.982 |
| SyntheticControl | 0.997 | **1.000** | 0.997 | 0.997 | 0.997 | 0.997 | 0.997 |
| ToeSegmentation1 | 0.974 | 0.965 | 0.969 | **0.974** | 0.965 | 0.969 | 0.952 |
| ToeSegmentation2 | 0.938 | 0.938 | **0.962** | **0.962** | 0.946 | 0.915 | 0.908 |
| Trace | **1.000** | **1.000** | **1.000** | **1.000** | **1.000** | **1.000** | **1.000** |
| TwoLeadECG | **1.000** | 0.999 | 0.994 | 0.998 | 0.996 | 0.999 | 0.998 |
| TwoPatterns | **1.000** | **1.000** | **1.000** | 0.999 | **1.000** | **1.000** | **1.000** |
| UMD | **1.000** | 0.993 | 0.986 | 0.979 | 0.986 | 0.993 | 0.993 |
| UWaveGestureLibraryAll | 0.975 | 0.974 | 0.970 | 0.968 | 0.951 | 0.975 | **0.980** |
| UWaveGestureLibraryX | **0.877** | 0.855 | 0.843 | 0.829 | 0.822 | 0.858 | 0.870 |
| UWaveGestureLibraryY | **0.823** | 0.775 | 0.771 | 0.747 | 0.771 | 0.773 | 0.802 |
| UWaveGestureLibraryZ | **0.845** | 0.797 | 0.786 | 0.777 | 0.769 | 0.794 | 0.816 |
| Wafer | 0.999 | **1.000** | 0.999 | 1.000 | 0.999 | 0.999 | 1.000 |
| Wine | 0.778 | 0.870 | 0.870 | 0.796 | 0.630 | 0.815 | **0.907** |
| WordSynonyms | 0.746 | 0.752 | **0.795** | 0.697 | 0.745 | 0.751 | 0.771 |
| Worms | 0.818 | 0.753 | **0.818** | 0.649 | 0.805 | 0.714 | 0.792 |
| WormsTwoClass | **0.844** | 0.792 | 0.831 | 0.766 | 0.766 | 0.740 | 0.792 |
| Yoga | 0.918 | **0.934** | 0.854 | 0.912 | 0.905 | 0.917 | 0.920 |
| Average | **0.884** | 0.877 | 0.865 | 0.865 | 0.858 | 0.853 | 0.868 |

Table 10: Full results for 125 UCR datasets of NuTime and self-supervised counterparts.

| Dataset | NuTime | TS2Vec-4 | TS2Vec-8 | TS2Vec-16 | T-Loss | TNC | TS-TCC | TST | DTW |
|---|---|---|---|---|---|---|---|---|---|
| ACSF1 | 0.678 | 0.840 | 0.900 | **0.910** | 0.900 | 0.730 | 0.730 | 0.760 | 0.640 |
| Adiac | **0.830** | 0.775 | 0.762 | 0.765 | 0.675 | 0.726 | 0.767 | 0.550 | 0.604 |
| AllGestureWiimoteX | 0.766 | 0.744 | **0.777** | 0.751 | 0.763 | 0.703 | 0.697 | 0.259 | 0.716 |
| AllGestureWiimoteY | 0.781 | 0.764 | **0.793** | 0.774 | 0.726 | 0.699 | 0.741 | 0.423 | 0.729 |
| AllGestureWiimoteZ | **0.776** | 0.734 | 0.746 | 0.770 | 0.723 | 0.646 | 0.689 | 0.447 | 0.643 |
| ArrowHead | **0.873** | 0.823 | 0.857 | 0.817 | 0.766 | 0.703 | 0.737 | 0.771 | 0.703 |
| BME | **1.000** | 0.973 | 0.993 | 0.980 | 0.993 | 0.973 | 0.933 | 0.760 | 0.900 |
| Beef | **0.887** | 0.767 | 0.767 | 0.633 | 0.667 | 0.733 | 0.600 | 0.500 | 0.633 |
| BeetleFly | 0.930 | 0.850 | 0.900 | 0.900 | 0.800 | 0.850 | 0.800 | **1.000** | 0.700 |
| BirdChicken | **1.000** | 0.800 | 0.800 | 0.800 | 0.850 | 0.750 | 0.650 | 0.650 | 0.750 |
| CBF | 0.995 | **1.000** | **1.000** | **1.000** | 0.983 | 0.983 | 0.998 | 0.898 | 0.997 |
| Car | **0.913** | 0.883 | 0.833 | 0.700 | 0.833 | 0.683 | 0.583 | 0.550 | 0.733 |
| Chinatown | 0.976 | 0.968 | 0.965 | 0.959 | 0.951 | 0.977 | **0.983** | 0.936 | 0.957 |
| ChlorineConcentration | 0.699 | 0.810 | **0.832** | 0.812 | 0.749 | 0.760 | 0.753 | 0.562 | 0.648 |
| CinCECGTorso | **0.919** | 0.812 | 0.827 | 0.825 | 0.713 | 0.669 | 0.671 | 0.508 | 0.651 |
| Coffee | **1.000** | **1.000** | **1.000** | **1.000** | **1.000** | **1.000** | **1.000** | 0.821 | **1.000** |
| Computers | **0.831** | 0.636 | 0.660 | 0.660 | 0.664 | 0.684 | 0.704 | 0.696 | 0.700 |
| CricketX | **0.815** | 0.800 | 0.782 | 0.805 | 0.713 | 0.623 | 0.731 | 0.385 | 0.754 |
| CricketY | **0.815** | 0.756 | 0.749 | 0.769 | 0.728 | 0.597 | 0.718 | 0.467 | 0.744 |
| CricketZ | **0.847** | 0.785 | 0.792 | 0.790 | 0.708 | 0.682 | 0.713 | 0.403 | 0.754 |
| Crop | 0.754 | 0.753 | **0.756** | 0.753 | 0.722 | 0.738 | 0.742 | 0.710 | 0.665 |
| DiatomSizeReduction | **0.994** | 0.980 | 0.984 | 0.987 | 0.984 | 0.993 | 0.977 | 0.961 | 0.967 |
| DistalPhalanxOutlineAgeGroup | **0.776** | 0.719 | 0.727 | 0.719 | 0.727 | 0.741 | 0.755 | 0.741 | 0.770 |
| DistalPhalanxOutlineCorrect | **0.811** | 0.775 | 0.761 | 0.757 | 0.775 | 0.754 | 0.754 | 0.728 | 0.717 |
| DistalPhalanxTW | **0.734** | 0.662 | 0.698 | 0.683 | 0.676 | 0.669 | 0.676 | 0.568 | 0.590 |
| ECG200 | 0.888 | 0.890 | 0.920 | 0.880 | **0.940** | 0.830 | 0.880 | 0.830 | 0.770 |
| ECG5000 | 0.938 | 0.935 | 0.935 | 0.934 | 0.933 | 0.937 | **0.941** | 0.928 | 0.924 |
| ECGFiveDays | 0.973 | **1.000** | **1.000** | **1.000** | **1.000** | 0.999 | 0.878 | 0.763 | 0.768 |
| EOGHorizontalSignal | **0.708** | 0.544 | 0.539 | 0.522 | 0.605 | 0.442 | 0.401 | 0.373 | 0.503 |
| EOGVerticalSignal | **0.591** | 0.467 | 0.503 | 0.472 | 0.434 | 0.392 | 0.376 | 0.298 | 0.448 |
| Earthquakes | **0.748** | 0.748 | 0.748 | 0.748 | 0.748 | 0.748 | 0.748 | 0.748 | 0.719 |
| ElectricDevices | **0.793** | 0.712 | 0.721 | 0.719 | 0.707 | 0.700 | 0.686 | 0.676 | 0.602 |
| EthanolLevel | **0.698** | 0.480 | 0.468 | 0.484 | 0.382 | 0.424 | 0.486 | 0.260 | 0.276 |
| FaceAll | **0.829** | 0.759 | 0.771 | 0.805 | 0.786 | 0.766 | 0.813 | 0.504 | 0.808 |
| FaceFour | **0.970** | 0.864 | 0.932 | 0.932 | 0.920 | 0.659 | 0.773 | 0.511 | 0.830 |
| FacesUCR | 0.919 | **0.930** | 0.924 | 0.926 | 0.884 | 0.789 | 0.863 | 0.543 | 0.905 |
| FiftyWords | **0.834** | 0.771 | 0.771 | 0.774 | 0.732 | 0.653 | 0.653 | 0.525 | 0.690 |
| Fish | **0.978** | 0.937 | 0.926 | 0.937 | 0.891 | 0.817 | 0.817 | 0.720 | 0.823 |
| FordA | 0.934 | 0.940 | 0.936 | **0.948** | 0.928 | 0.902 | 0.930 | 0.568 | 0.555 |
| FordB | 0.799 | 0.789 | 0.794 | 0.807 | 0.793 | 0.733 | **0.815** | 0.507 | 0.620 |
| FreezerRegularTrain | **0.999** | 0.985 | 0.986 | 0.983 | 0.956 | 0.991 | 0.989 | 0.922 | 0.899 |
| FreezerSmallTrain | **0.985** | 0.894 | 0.870 | 0.872 | 0.933 | 0.982 | 0.979 | 0.920 | 0.753 |
| Fungi | 0.881 | 0.962 | 0.957 | 0.946 | **1.000** | 0.527 | 0.753 | 0.366 | 0.839 |
| GestureMidAirD1 | **0.774** | 0.631 | 0.608 | 0.615 | 0.608 | 0.431 | 0.369 | 0.208 | 0.569 |
| GestureMidAirD2 | **0.700** | 0.508 | 0.469 | 0.515 | 0.546 | 0.362 | 0.254 | 0.138 | 0.608 |
| GestureMidAirD3 | **0.532** | 0.346 | 0.292 | 0.300 | 0.285 | 0.292 | 0.177 | 0.154 | 0.323 |
| GesturePebbleZ1 | **0.951** | 0.878 | 0.930 | 0.884 | 0.919 | 0.378 | 0.395 | 0.500 | 0.791 |

| | | | | | | | | | |
|---|---|---|---|---|---|---|---|---|---|
| GesturePebbleZ2 | 0.895 | 0.842 | 0.873 | 0.848 | **0.899** | 0.316 | 0.430 | 0.380 | 0.671 |
| GunPoint | **0.997** | 0.980 | 0.980 | 0.987 | 0.980 | 0.967 | 0.993 | 0.827 | 0.907 |
| GunPointAgeSpan | **0.998** | 0.994 | 0.987 | 0.968 | 0.994 | 0.984 | 0.994 | 0.991 | 0.918 |
| GunPointMaleVersusFemale | **1.000** | **1.000** | **1.000** | **1.000** | 0.997 | 0.994 | 0.997 | **1.000** | 0.997 |
| GunPointOldVersusYoung | **1.000** | **1.000** | **1.000** | **1.000** | **1.000** | **1.000** | **1.000** | **1.000** | 0.838 |
| Ham | **0.762** | 0.714 | 0.714 | 0.724 | 0.724 | 0.752 | 0.743 | 0.524 | 0.467 |
| HandOutlines | **0.963** | 0.919 | 0.922 | 0.930 | 0.922 | 0.930 | 0.724 | 0.735 | 0.881 |
| Haptics | **0.551** | 0.510 | 0.526 | 0.536 | 0.490 | 0.474 | 0.396 | 0.357 | 0.377 |
| Herring | **0.694** | 0.625 | 0.641 | 0.609 | 0.594 | 0.594 | 0.594 | 0.594 | 0.531 |
| HouseTwenty | **0.958** | 0.941 | 0.916 | 0.941 | 0.933 | 0.782 | 0.790 | 0.815 | 0.924 |
| InlineSkate | **0.423** | 0.389 | 0.415 | 0.407 | 0.371 | 0.378 | 0.347 | 0.287 | 0.384 |
| InsectEPGRegularTrain | **1.000** | **1.000** | **1.000** | **1.000** | **1.000** | **1.000** | **1.000** | **1.000** | 0.872 |
| InsectEPGSmallTrain | **1.000** | **1.000** | **1.000** | **1.000** | **1.000** | **1.000** | **1.000** | **1.000** | 0.735 |
| InsectWingbeatSound | 0.594 | 0.629 | **0.630** | 0.624 | 0.597 | 0.549 | 0.415 | 0.266 | 0.355 |
| ItalyPowerDemand | 0.949 | **0.961** | 0.925 | 0.960 | 0.954 | 0.928 | 0.955 | 0.845 | 0.950 |
| LargeKitchenAppliances | **0.882** | 0.845 | 0.845 | 0.875 | 0.789 | 0.776 | 0.848 | 0.595 | 0.795 |
| Lightning2 | 0.830 | 0.836 | **0.869** | 0.820 | **0.869** | **0.869** | 0.836 | 0.705 | **0.869** |
| Lightning7 | 0.833 | 0.836 | **0.863** | 0.822 | 0.795 | 0.767 | 0.685 | 0.411 | 0.726 |
| Mallat | **0.965** | 0.915 | 0.914 | 0.873 | 0.951 | 0.871 | 0.922 | 0.713 | 0.934 |
| Meat | **0.987** | 0.950 | 0.950 | 0.967 | 0.950 | 0.917 | 0.883 | 0.900 | 0.933 |
| MedicalImages | **0.801** | 0.792 | 0.789 | 0.793 | 0.750 | 0.754 | 0.747 | 0.632 | 0.737 |
| MelbournePedestrian | **0.966** | 0.954 | 0.959 | 0.956 | 0.944 | 0.942 | 0.949 | 0.741 | 0.791 |
| MiddlePhalanxOutlineAgeGroup | 0.630 | 0.636 | 0.636 | 0.630 | **0.656** | 0.643 | 0.630 | 0.617 | 0.500 |
| MiddlePhalanxOutlineCorrect | **0.859** | 0.811 | 0.838 | 0.825 | 0.825 | 0.818 | 0.818 | 0.753 | 0.698 |
| MiddlePhalanxTW | 0.587 | 0.591 | 0.584 | 0.578 | 0.591 | 0.571 | **0.610** | 0.506 | 0.506 |
| MixedShapesRegularTrain | **0.975** | 0.915 | 0.917 | 0.922 | 0.905 | 0.911 | 0.855 | 0.879 | 0.842 |
| MixedShapesSmallTrain | **0.942** | 0.881 | 0.861 | 0.856 | 0.860 | 0.813 | 0.735 | 0.828 | 0.780 |
| MoteStrain | **0.964** | 0.857 | 0.861 | 0.863 | 0.851 | 0.825 | 0.843 | 0.768 | 0.835 |
| NonInvasiveFetalECGThorax1 | 0.919 | 0.923 | **0.930** | 0.919 | 0.878 | 0.898 | 0.898 | 0.471 | 0.790 |
| NonInvasiveFetalECGThorax2 | 0.928 | **0.940** | 0.938 | 0.935 | 0.919 | 0.912 | 0.913 | 0.832 | 0.865 |
| OSULeaf | **0.945** | 0.876 | 0.851 | 0.843 | 0.760 | 0.723 | 0.723 | 0.545 | 0.591 |
| OliveOil | 0.813 | **0.900** | **0.900** | **0.900** | 0.867 | 0.833 | 0.800 | 0.800 | 0.833 |
| PLAID | **0.945** | 0.551 | 0.561 | 0.549 | 0.555 | 0.495 | 0.445 | 0.419 | 0.840 |
| PhalangesOutlinesCorrect | **0.850** | 0.795 | 0.809 | 0.823 | 0.784 | 0.787 | 0.804 | 0.773 | 0.728 |
| Phoneme | **0.349** | 0.296 | 0.312 | 0.309 | 0.276 | 0.180 | 0.242 | 0.139 | 0.228 |
| PickupGestureWiimoteZ | **0.848** | 0.800 | 0.820 | 0.760 | 0.740 | 0.620 | 0.600 | 0.240 | 0.660 |
| PigAirwayPressure | **0.862** | 0.524 | 0.630 | 0.683 | 0.510 | 0.413 | 0.380 | 0.120 | 0.106 |
| PigArtPressure | **0.994** | 0.962 | 0.966 | 0.966 | 0.928 | 0.808 | 0.524 | 0.774 | 0.245 |
| PigCVP | **0.947** | 0.803 | 0.812 | 0.870 | 0.788 | 0.649 | 0.615 | 0.596 | 0.154 |
| Plane | **1.000** | **1.000** | **1.000** | 0.990 | 0.990 | **1.000** | **1.000** | 0.933 | **1.000** |
| PowerCons | **0.992** | 0.967 | 0.961 | 0.972 | 0.900 | 0.933 | 0.961 | 0.911 | 0.878 |
| ProximalPhalanxOutlineAgeGroup | **0.885** | 0.844 | 0.834 | 0.829 | 0.844 | 0.854 | 0.839 | 0.854 | 0.805 |
| ProximalPhalanxOutlineCorrect | **0.931** | 0.876 | 0.887 | 0.900 | 0.859 | 0.866 | 0.873 | 0.770 | 0.784 |
| ProximalPhalanxTW | **0.831** | 0.785 | 0.824 | 0.805 | 0.771 | 0.810 | 0.800 | 0.780 | 0.761 |
| RefrigerationDevices | **0.643** | 0.587 | 0.589 | 0.589 | 0.515 | 0.565 | 0.563 | 0.483 | 0.464 |
| Rock | **0.896** | 0.660 | 0.700 | 0.700 | 0.580 | 0.580 | 0.600 | 0.680 | 0.600 |
| ScreenType | **0.601** | 0.405 | 0.411 | 0.397 | 0.416 | 0.509 | 0.419 | 0.419 | 0.397 |
| SemgHandGenderCh2 | 0.946 | 0.952 | **0.963** | 0.962 | 0.890 | 0.882 | 0.837 | 0.725 | 0.802 |
| SemgHandMovementCh2 | 0.863 | **0.893** | 0.860 | 0.891 | 0.789 | 0.593 | 0.613 | 0.420 | 0.584 |
| SemgHandSubjectCh2 | 0.921 | 0.944 | **0.951** | 0.942 | 0.853 | 0.771 | 0.753 | 0.484 | 0.727 |
| ShakeGestureWiimoteZ | **0.948** | 0.940 | 0.940 | 0.920 | 0.920 | 0.820 | 0.860 | 0.760 | 0.860 |
| ShapeletSim | **1.000** | 0.989 | **1.000** | 0.994 | 0.672 | 0.589 | 0.683 | 0.489 | 0.650 |
| ShapesAll | **0.927** | 0.897 | 0.902 | 0.905 | 0.848 | 0.788 | 0.773 | 0.733 | 0.768 |
| SmallKitchenAppliances | **0.851** | 0.723 | 0.731 | 0.733 | 0.677 | 0.725 | 0.691 | 0.592 | 0.643 |
| SmoothSubspace | 0.991 | 0.967 | 0.980 | **0.993** | 0.960 | 0.913 | 0.953 | 0.827 | 0.827 |
| SonyAIBORobotSurface1 | **0.948** | 0.874 | 0.903 | 0.900 | 0.902 | 0.804 | 0.899 | 0.724 | 0.725 |
| SonyAIBORobotSurface2 | **0.916** | 0.890 | 0.871 | 0.889 | 0.889 | 0.834 | 0.907 | 0.745 | 0.831 |
| StarLightCurves | **0.980** | 0.970 | 0.969 | 0.971 | 0.964 | 0.968 | 0.967 | 0.949 | 0.907 |
| Strawberry | **0.972** | 0.962 | 0.962 | 0.965 | 0.954 | 0.951 | 0.965 | 0.916 | 0.941 |
| SwedishLeaf | **0.965** | 0.939 | 0.941 | 0.942 | 0.914 | 0.880 | 0.923 | 0.738 | 0.792 |
| Symbols | **0.982** | 0.973 | 0.976 | 0.972 | 0.963 | 0.885 | 0.916 | 0.786 | 0.950 |
| SyntheticControl | 0.996 | 0.997 | 0.997 | 0.993 | 0.987 | **1.000** | 0.990 | 0.490 | 0.993 |
| ToeSegmentation1 | **0.961** | 0.930 | 0.917 | 0.947 | 0.939 | 0.864 | 0.930 | 0.807 | 0.772 |
| ToeSegmentation2 | **0.946** | 0.915 | 0.892 | 0.900 | 0.900 | 0.831 | 0.877 | 0.615 | 0.838 |
| Trace | **1.000** | **1.000** | **1.000** | **1.000** | 0.990 | **1.000** | **1.000** | **1.000** | **1.000** |
| TwoLeadECG | 0.999 | 0.982 | 0.986 | 0.987 | **0.999** | 0.993 | 0.976 | 0.871 | 0.905 |
| TwoPatterns | **1.000** | **1.000** | **1.000** | **1.000** | 0.999 | **1.000** | 0.999 | 0.466 | **1.000** |
| UMD | **1.000** | **1.000** | **1.000** | 0.993 | 0.993 | 0.993 | 0.986 | 0.910 | 0.993 |
| UWaveGestureLibraryAll | **0.971** | 0.934 | 0.930 | 0.934 | 0.896 | 0.903 | 0.692 | 0.475 | 0.892 |
| UWaveGestureLibraryX | **0.859** | 0.810 | 0.795 | 0.801 | 0.785 | 0.781 | 0.733 | 0.569 | 0.728 |
| UWaveGestureLibraryY | **0.800** | 0.729 | 0.719 | 0.720 | 0.710 | 0.697 | 0.641 | 0.348 | 0.634 |
| UWaveGestureLibraryZ | **0.821** | 0.761 | 0.770 | 0.768 | 0.757 | 0.721 | 0.690 | 0.655 | 0.658 |
| Wafer | 0.999 | 0.995 | 0.998 | 0.997 | 0.992 | 0.994 | 0.994 | 0.991 | 0.980 |
| Wine | 0.785 | 0.778 | 0.870 | **0.889** | 0.815 | 0.759 | 0.778 | 0.500 | 0.574 |
| WordSynonyms | **0.712** | 0.699 | 0.676 | 0.704 | 0.691 | 0.630 | 0.531 | 0.422 | 0.649 |
| Worms | **0.831** | 0.701 | 0.701 | 0.701 | 0.727 | 0.623 | 0.753 | 0.455 | 0.584 |
| WormsTwoClass | **0.839** | 0.805 | 0.805 | 0.753 | 0.792 | 0.727 | 0.753 | 0.584 | 0.623 |
| Yoga | **0.904** | 0.880 | 0.887 | 0.877 | 0.837 | 0.812 | 0.791 | 0.830 | 0.837 |
| Average | **0.869** | 0.824 | 0.830 | 0.827 | 0.806 | 0.761 | 0.757 | 0.641 | 0.727 |

Table 11: Full results for 26 UEA datasets of NuTime and supervised counterparts.

| Dataset | NuTime | HC2 | ROCKET | Arsenal | DrCIF | CIF | HC1 | TDE | STC | DTW-C |
|---|---|---|---|---|---|---|---|---|---|---|
| ArticularyWordRecognition | 0.993 | 0.993 | **0.997** | 0.993 | 0.980 | 0.983 | 0.990 | 0.993 | 0.990 | 0.987 |
| AtrialFibrillation | 0.200 | 0.267 | 0.200 | 0.133 | **0.333** | **0.333** | 0.133 | 0.267 | 0.267 | 0.200 |
| BasicMotions | **1.000** | **1.000** | **1.000** | **1.000** | **1.000** | **1.000** | **1.000** | **1.000** | 0.975 | 0.975 |
| Cricket | **1.000** | **1.000** | **1.000** | **1.000** | 0.986 | 0.986 | 0.986 | 0.986 | 0.986 | **1.000** |
| DuckDuckGeese | 0.540 | 0.560 | 0.500 | 0.460 | 0.540 | 0.440 | 0.480 | 0.340 | 0.360 | **0.580** |
| ERing | 0.993 | 0.989 | 0.985 | 0.981 | **0.993** | 0.981 | 0.970 | 0.963 | 0.889 | 0.915 |
| EigenWorms | 0.916 | **0.947** | 0.908 | 0.901 | 0.924 | 0.916 | 0.634 | 0.939 | 0.779 | 0.618 |
| Epilepsy | 0.894 | **1.000** | 0.978 | 0.986 | 0.978 | 0.986 | **1.000** | 0.993 | 0.993 | 0.964 |
| EthanolConcentration | 0.494 | 0.772 | 0.445 | 0.460 | 0.692 | 0.734 | 0.791 | 0.555 | **0.821** | 0.323 |
| FaceDetection | **0.684** | 0.660 | 0.648 | 0.653 | 0.620 | 0.627 | 0.656 | 0.564 | 0.646 | 0.529 |
| FingerMovements | **0.630** | 0.530 | 0.540 | 0.530 | 0.600 | 0.520 | 0.550 | 0.560 | 0.510 | 0.530 |
| HandMovementDirection | 0.527 | 0.473 | 0.459 | 0.473 | 0.527 | **0.595** | 0.446 | 0.378 | 0.392 | 0.189 |
| Handwriting | 0.229 | 0.548 | 0.567 | 0.541 | 0.346 | 0.356 | 0.482 | 0.561 | 0.288 | **0.607** |
| Heartbeat | 0.785 | 0.732 | 0.741 | 0.741 | **0.790** | 0.780 | 0.722 | 0.746 | 0.722 | 0.717 |
| LSST | **0.721** | 0.643 | 0.622 | 0.642 | 0.556 | 0.573 | 0.575 | 0.570 | 0.587 | 0.551 |
| Libras | **0.967** | 0.933 | 0.900 | 0.906 | 0.894 | 0.911 | 0.900 | 0.850 | 0.861 | 0.872 |
| MotorImagery | **0.630** | 0.540 | 0.510 | 0.530 | 0.440 | 0.500 | 0.610 | 0.590 | 0.500 | 0.500 |
| NATOPS | **0.944** | 0.894 | 0.883 | 0.883 | 0.844 | 0.856 | 0.889 | 0.839 | 0.872 | 0.883 |
| PEMS-SF | 0.931 | **1.000** | 0.821 | 0.827 | **1.000** | **1.000** | 0.983 | **1.000** | 0.971 | 0.711 |
| PenDigits | **0.993** | 0.979 | 0.985 | 0.983 | 0.977 | 0.967 | 0.934 | 0.935 | 0.941 | 0.977 |
| PhonemeSpectra | **0.349** | 0.290 | 0.259 | 0.277 | 0.308 | 0.265 | 0.321 | 0.246 | 0.295 | 0.151 |
| RacketSports | **0.941** | 0.908 | 0.914 | 0.901 | 0.901 | 0.882 | 0.888 | 0.836 | 0.888 | 0.803 |
| SelfRegulationSCP1 | 0.887 | **0.891** | 0.843 | 0.846 | 0.877 | 0.860 | 0.853 | 0.812 | 0.840 | 0.775 |
| SelfRegulationSCP2 | **0.611** | 0.500 | 0.494 | 0.494 | 0.494 | 0.500 | 0.461 | 0.500 | 0.533 | 0.539 |
| StandWalkJump | **0.733** | 0.467 | 0.600 | 0.533 | 0.533 | 0.400 | 0.333 | 0.333 | 0.467 | 0.200 |
| UWaveGestureLibrary | **0.963** | 0.928 | 0.931 | 0.928 | 0.909 | 0.925 | 0.891 | 0.925 | 0.850 | 0.903 |
| Average | **0.752** | 0.748 | 0.721 | 0.716 | 0.732 | 0.726 | 0.711 | 0.703 | 0.701 | 0.654 |

## D.2 Multivariate Time Series Classification

The full results, in terms of top-1 accuracy, for both supervised and self-supervised pretrained methods on each sub-dataset in the UEA benchmark are presented in Table 11 and Table 12 respectively.

## D.3 Few-shot Learning

Figure 9 shows the plot accuracy comparison between the proposed NuTime and a sophisticated meta-learning approach (Narwariya et al., 2020). Additionally, Table 13 provides the full classification results on 41 datasets from the UCR archive for few-shot learning.

## E   Attention Mechanism Visualization in NuTime

We provide the visualizations for attention scores between the [CLS] token and other patch tokens of the input series in the last layer of NuTime in Figure 10 and 11 to help interpret the learned representations.

Table 12: Full results for 29 UEA datasets of NuTime and self-supervised counterparts.

| Dataset | NuTime | TS2Vec | T-Loss | TNC | TS-TCC | TST | DTW |
|---|---|---|---|---|---|---|---|
| ArticularyWordRecognition | **0.994** | 0.987 | 0.943 | 0.973 | 0.953 | 0.977 | 0.987 |
| AtrialFibrillation | **0.347** | 0.200 | 0.133 | 0.133 | 0.267 | 0.067 | 0.200 |
| BasicMotions | **1.000** | 0.975 | **1.000** | 0.975 | **1.000** | 0.975 | 0.975 |
| CharacterTrajectories | 0.994 | **0.995** | 0.993 | 0.967 | 0.985 | 0.975 | 0.989 |
| Cricket | **1.000** | 0.972 | 0.972 | 0.958 | 0.917 | **1.000** | **1.000** |
| DuckDuckGeese | 0.552 | **0.680** | 0.650 | 0.460 | 0.380 | 0.620 | 0.600 |
| ERing | **0.986** | 0.874 | 0.133 | 0.852 | 0.904 | 0.874 | 0.133 |
| EigenWorms | **0.910** | 0.847 | 0.840 | 0.840 | 0.779 | 0.748 | 0.618 |
| Epilepsy | **0.993** | 0.964 | 0.971 | 0.957 | 0.957 | 0.949 | 0.964 |
| EthanolConcentration | **0.466** | 0.308 | 0.205 | 0.297 | 0.285 | 0.262 | 0.323 |
| FaceDetection | **0.663** | 0.501 | 0.513 | 0.536 | 0.544 | 0.534 | 0.529 |
| FingerMovements | **0.612** | 0.480 | 0.580 | 0.470 | 0.460 | 0.560 | 0.530 |
| HandMovementDirection | **0.532** | 0.338 | 0.351 | 0.324 | 0.243 | 0.243 | 0.231 |
| Handwriting | 0.228 | **0.515** | 0.451 | 0.249 | 0.498 | 0.225 | 0.286 |
| Heartbeat | **0.784** | 0.683 | 0.741 | 0.746 | 0.751 | 0.746 | 0.717 |
| JapaneseVowels | 0.983 | 0.984 | **0.989** | 0.978 | 0.930 | 0.978 | 0.949 |
| LSST | **0.693** | 0.537 | 0.509 | 0.595 | 0.474 | 0.408 | 0.551 |
| Libras | **0.976** | 0.867 | 0.883 | 0.817 | 0.822 | 0.656 | 0.870 |
| MotorImagery | **0.622** | 0.510 | 0.580 | 0.500 | 0.610 | 0.500 | 0.500 |
| NATOPS | **0.940** | 0.928 | 0.917 | 0.911 | 0.822 | 0.850 | 0.883 |
| PEMS-SF | **0.925** | 0.682 | 0.676 | 0.699 | 0.734 | 0.740 | 0.711 |
| PenDigits | 0.988 | **0.989** | 0.981 | 0.979 | 0.974 | 0.560 | 0.977 |
| PhonemeSpectra | **0.320** | 0.233 | 0.222 | 0.207 | 0.252 | 0.085 | 0.151 |
| RacketSports | **0.934** | 0.855 | 0.855 | 0.776 | 0.816 | 0.809 | 0.803 |
| SelfRegulationSCP1 | **0.899** | 0.812 | 0.843 | 0.799 | 0.823 | 0.754 | 0.775 |
| SelfRegulationSCP2 | **0.603** | 0.578 | 0.539 | 0.550 | 0.533 | 0.550 | 0.539 |
| SpokenArabicDigits | **0.993** | 0.988 | 0.905 | 0.934 | 0.970 | 0.923 | 0.963 |
| StandWalkJump | **0.667** | 0.467 | 0.333 | 0.400 | 0.333 | 0.267 | 0.200 |
| UWaveGestureLibrary | **0.955** | 0.906 | 0.875 | 0.759 | 0.753 | 0.575 | 0.903 |
| Average | **0.778** | 0.712 | 0.675 | 0.677 | 0.682 | 0.635 | 0.650 |

Table 13: Full results for few-shot learning on 41 UCR datasets.

| Dataset | NuTime | FS-1 | FS-2 | ResNet | BOSS | DTW | ED |
|---|---|---|---|---|---|---|---|
| Adiac | **0.781** | 0.671 | 0.674 | 0.539 | 0.709 | 0.540 | 0.538 |
| Beef | **0.716** | 0.653 | 0.595 | 0.519 | 0.701 | 0.626 | 0.618 |
| BeetleFly | 0.809 | 0.900 | **0.958** | 0.702 | 0.789 | 0.614 | 0.667 |
| BirdChicken | **1.000** | 0.929 | 1.000 | 0.692 | 0.921 | 0.496 | 0.468 |
| ChlorineConcentration | **0.502** | 0.329 | 0.331 | 0.342 | 0.356 | 0.338 | 0.339 |
| Coffee | 0.909 | **0.978** | 0.970 | 0.934 | 0.977 | 0.914 | 0.920 |
| CricketX | 0.550 | **0.594** | 0.544 | 0.555 | 0.491 | 0.567 | 0.348 |
| CricketY | 0.525 | **0.562** | 0.516 | 0.505 | 0.461 | 0.556 | 0.375 |
| CricketZ | 0.530 | **0.598** | 0.541 | 0.523 | 0.481 | 0.560 | 0.357 |
| DistalPhalanxOutlineAgeGroup | **0.738** | 0.664 | 0.705 | 0.709 | 0.658 | 0.698 | 0.710 |
| DistalPhalanxOutlineCorrec | **0.643** | 0.588 | 0.569 | 0.609 | 0.575 | 0.583 | 0.571 |
| DistalPhalanxTW | **0.508** | 0.463 | 0.481 | 0.476 | 0.437 | 0.448 | 0.444 |
| ECG200 | 0.699 | 0.758 | 0.738 | 0.712 | 0.728 | 0.755 | **0.771** |
| ECG5000 | 0.537 | 0.533 | **0.548** | 0.533 | 0.533 | 0.494 | 0.524 |
| ECGFiveDays | 0.781 | **0.939** | 0.928 | 0.916 | 0.909 | 0.666 | 0.685 |
| ElectricDevices | **0.569** | 0.375 | 0.380 | 0.381 | 0.351 | 0.423 | 0.239 |
| FaceAll | 0.789 | 0.785 | 0.712 | 0.742 | **0.795** | 0.764 | 0.545 |
| FaceFour | 0.932 | 0.934 | 0.958 | 0.792 | **1.000** | 0.869 | 0.812 |
| FordA | 0.477 | **0.797** | 0.777 | 0.769 | 0.693 | 0.541 | 0.561 |
| FordB | 0.733 | **0.787** | 0.726 | 0.692 | 0.585 | 0.535 | 0.515 |
| InsectWingbeatSound | 0.396 | 0.487 | 0.452 | 0.485 | 0.398 | 0.473 | **0.489** |
| Meat | 0.885 | 0.890 | 0.880 | 0.559 | 0.876 | 0.919 | **0.919** |
| MedicalImages | 0.625 | 0.592 | 0.585 | 0.620 | 0.488 | **0.675** | 0.579 |
| MiddlePhalanxOutlineAgeGroup | 0.371 | 0.547 | 0.515 | 0.527 | 0.478 | **0.558** | 0.529 |
| MiddlePhalanxOutlineCorrect | **0.631** | 0.529 | 0.531 | 0.540 | 0.526 | 0.550 | 0.563 |
| MiddlePhalanxTW | **0.380** | 0.353 | 0.351 | 0.341 | 0.348 | 0.339 | 0.338 |
| PhalangesOutlinesCorrect | 0.526 | 0.539 | 0.536 | **0.544** | 0.512 | 0.535 | 0.532 |
| ProximalPhalanxOutlineAgeGroup | **0.813** | 0.682 | 0.697 | 0.729 | 0.731 | 0.719 | 0.692 |
| ProximalPhalanxOutlineCorrect | **0.710** | 0.634 | 0.638 | 0.650 | 0.645 | 0.626 | 0.633 |
| ProximalPhalanxTW | 0.471 | 0.432 | 0.411 | **0.517** | 0.419 | 0.445 | 0.427 |
| Strawberry | **0.777** | 0.741 | 0.755 | 0.722 | 0.714 | 0.671 | 0.682 |
| SwedishLeaf | **0.845** | 0.776 | 0.778 | 0.765 | 0.776 | 0.690 | 0.599 |
| SyntheticControl | 0.937 | **0.971** | 0.948 | 0.960 | 0.867 | 0.958 | 0.736 |
| TwoPatterns | 0.903 | 0.831 | 0.811 | 0.874 | 0.692 | **0.970** | 0.361 |
| UWaveGestureLibraryX | **0.753** | 0.606 | 0.546 | 0.598 | 0.479 | 0.615 | 0.591 |
| UWaveGestureLibraryY | **0.587** | 0.478 | 0.430 | 0.478 | 0.363 | 0.518 | 0.504 |
| UWaveGestureLibraryZ | 0.570 | **0.599** | 0.541 | 0.570 | 0.489 | 0.551 | 0.536 |
| Wafer | 0.900 | 0.892 | 0.894 | 0.911 | **0.936** | 0.922 | 0.922 |
| Wine | 0.556 | 0.578 | **0.631** | 0.562 | 0.571 | 0.493 | 0.496 |
| Yoga | **0.649** | 0.528 | 0.546 | 0.501 | 0.548 | 0.525 | 0.505 |
| Average | **0.675** | 0.663 | 0.653 | 0.627 | 0.625 | 0.618 | 0.566 |

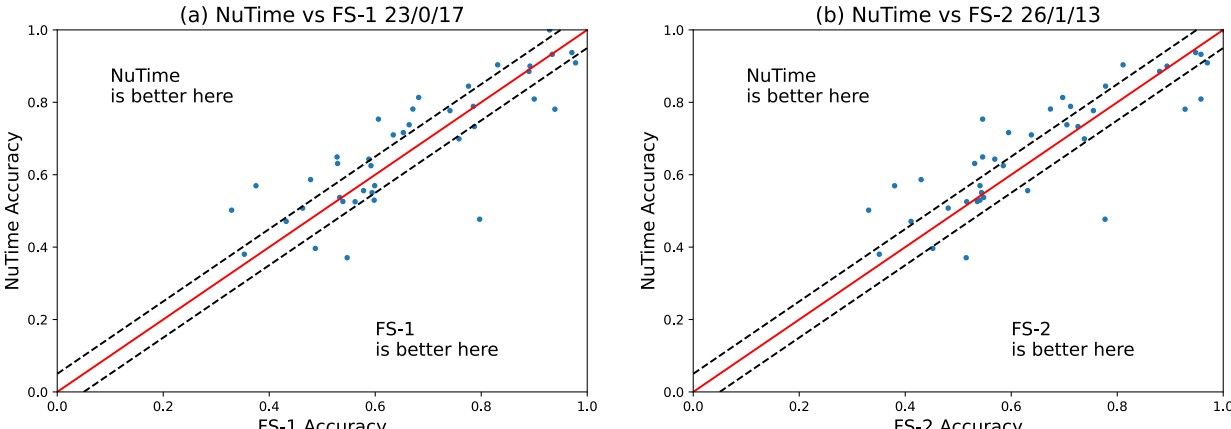

Figure 9: Plot accuracy comparison of NuTime and (a) FS-1, (b) FS-2 on 41 datasets from the UCR archive. Each subfigure's title displays the win/tie/loss comparison between NuTime and other methods. The two dotted lines indicate the 5% interval.

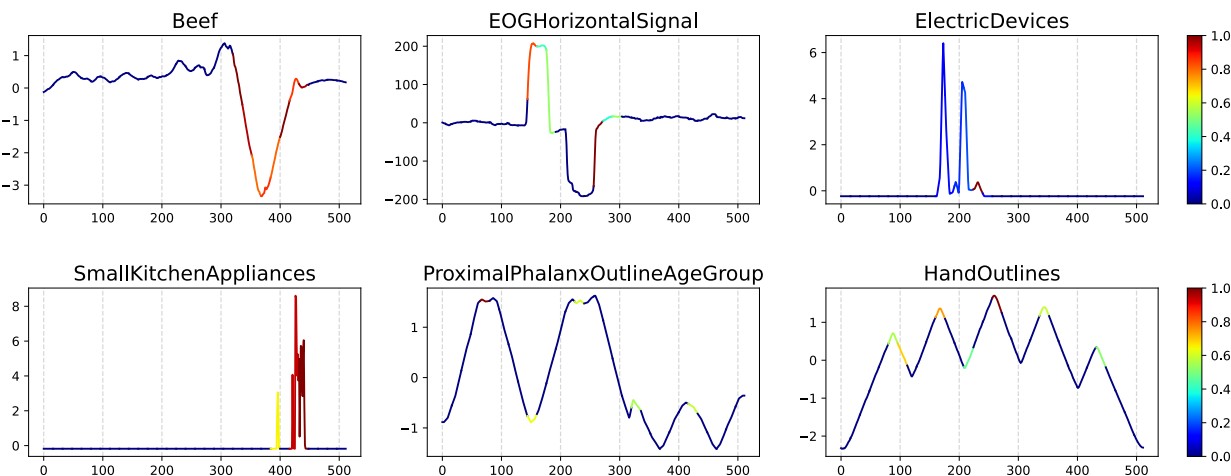

Figure 10: Attention scores between the `[CLS]` token and other patch tokens of the input series in the NuTime model last layer. A higher score indicates that the learned representation is more closely associated with those patches. Six datasets from the UCR archive are randomly selected for visualization.

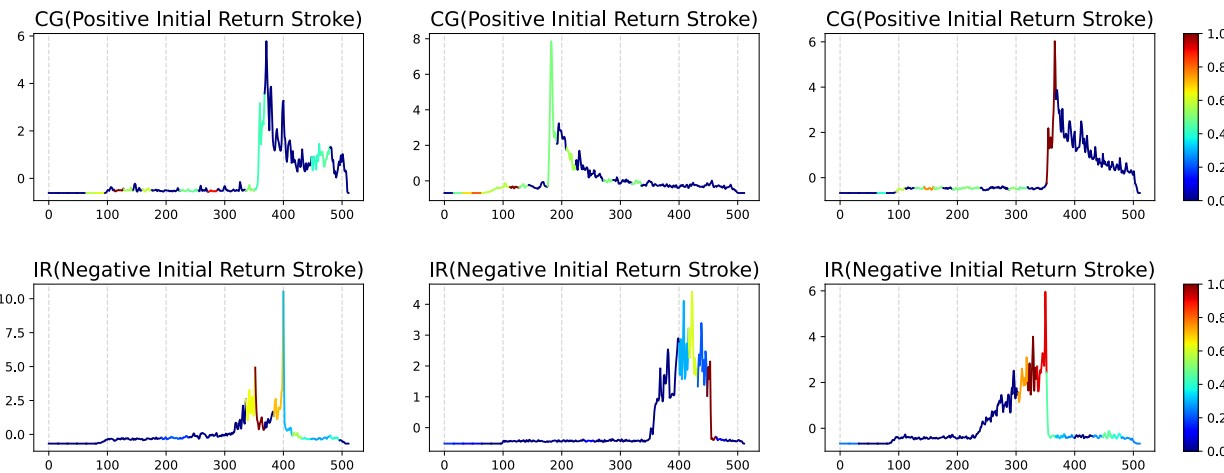

Figure 11: Attention scores between the [CLS] token and other patch tokens of the input series in the NuTime model last layer. These series are randomly selected from the LIGHTNING7 dataset. The time series are categorized into two classes: CG (Positive Initial Return Stroke) and IR (Negative Initial Return Stroke). CG-type time series in the first row are characterized by a sudden surge of radiation, followed by a brief period of noise. IR-type series in the second row slowly build up and then sharply decline in an exponential shape. The visualization implies that the NuTime learned representations which are helpful for classification.

