# OpenReview forum: "NuTime: Numerically Multi-Scaled Embedding for Large- Scale Time-Series Pretraining"
_TMLR — Accepted by TMLR_

### Review · Reviewer_T9zk · 2024-04-21

**Summary Of Contributions:**

This paper proposes a new self-supervised method for time series analysis, called NuTime.  It leverages a standard and generic transformer network, with the key addition being how the data is pre-processed before feeding it into the network.  It chunks the time series into non-overlapping chunks, and each chunks sends in a normalized shape (the chunk of values after standard (x-mean)/std normalization), a mean value, and a standard deviation.  This removes some scale notions from the "shape", but also passes it to the transformer network separately.

Within this, there is a parameter k, which is used to regularize the normalization (its square is added to the variance in the denominator).  This parameter seems to be needed, but can be set in a wide range.

The results beats state of the art on several benchmarks.

**Audience:**

Yes

**Claims And Evidence:**

Yes

**Requested Changes:**

###  Smaller writing suggestions and questions:

1. While the evaluation measures seem to be standard data sets and tasks, they are really treated as a black box in this paper.  It would have been useful to explain a bit about what sorts of things are being classified, and why classification is a useful task in these settings.

 1.  BYOL seems to be an integral enough part of your approach.  I suggest to elaborate just a bit on how it works, and perhaps what the acronym means (although I guess just knowing that expansion does not help the reader too much).

 2.  Why did you choose to use **non**-overlapping windows, instead of shingled windows?  This is worth addressing in the paper.

 3. How do you know you have right range and resolution on k  (in [10^-4, 10^-3, .., 1, .., 10^4])?  I see the ablation study, but based on Fig 3(a), it seems the effect can be pretty large between values of k that you tried.
Is k=0 represented here (is that what Num Scales = 0 means?).  Please expand on this a bit more.

**Strengths And Weaknesses:**

## Subjective evaluation:

 - I like the simple normalization idea + transformers.  Surprising it beats more engineered approaches.
 - I like the focus on normalization and pre-processing, as something that can be isolated, and still take advantage of big hammers like transformers.

## TMLR criteria evaluation:
Overall it approaches a problem of clear relevance to TMLR, and general importance.  The solution is clear, simple, and effective.  The methods are described clear enough, and the evaluation on standard benchmarks holds up, and shows state of the art performance.

---

> ### Author Response · Authors · 2024-05-06
> **Response to reviewer T9zk**
>
> We thank the reviewer for the valuable feedbacks. The requested changes are incorporated to the revised paper, and highlighted in blue.
>
> Below, we respond to the individual concerns.
>
> * Explain classification datasets used in the paper.
>
> The detailed information about the classification datasets is appended in Appendix A. For example, the SleepEEG recordings are classified into 5 patterns in including Wake (W), Non-rapid eye movement (N1, N2, N3), and Rapid Eye Movement (REM).
>
> * Elaborate more about BYOL.
>
> We thank the reviewer for the suggestion. We have updated the text in the paper.
>
> * Why non-overlapping windows instead of shingled windows.
>
> We choose to use non-overlapping windows mainly to resemble the use of non-overlapping patches in Vision Transformers. We have revised the main text to make this clear.
>
> * How to choose the range and the resolution of k and can k be set to 0.
>
> We chose the range of k wide enough to cover the mean and std statistics across the whole training dataset. The resolution of k, separated by a scale of 10, is determined empirically.  We have added this information to the main text.
> The value of k cannot be set to zero. According to the Eqn(2),  the embedding vector will collapse into a constant vector regardless of the input x if k is set to zero.

---

> > ### Comment · Reviewer_T9zk · 2024-05-06
> >
> > Thanks to the authors for their response.  My major questions have been addressed.

---

### Review · Reviewer_P6SZ · 2024-04-21

**Summary Of Contributions:**

The submission proposes a method for better encoding of time-series data for use in large scale pretraining. The method involves segment-wise normalization (mean subtraction and scaling by the std). To include the information lost via normalization, the authors propose to use a multiscale embedding of these scalar quantities.

**Audience:**

Yes

**Claims And Evidence:**

Yes

**Requested Changes:**

**On NME:** The author's proposal for the embedding layer is interesting. But the author's ablations do not answer the question of whether or not this is over-engineered. In Eq. 2 the authors show that the effect is very similar to just $\frac{x}{\sqrt{c_1 * x^2 + c_2^2}}$. Instead of relying on layer norm and w and b, why not just use a number of these (either randomly samples c_1 and c_2, appropriately in log space or handpicked) as your embedding?
- Perform ablation tests that show that the complexity of NME is necessary. Or show that NME can be simplified. The ablations that I can think of are 1. y's as described as above but still using the weighted average. 2. Replace weighted averaging with concatenation (just pick a smaller y_i subspace).
- In Fig. 3, it seems like the layer norm is shared between all z_i, but Eq. 2 implies each is normalized separately which makes more sense. I would suggest breaking up the layer norm block so that it is clear that each z is normalized individually.
- I like Fig. 3a. It would be instructive to see the same figure but showing a few channels of the scalar value embedding e(x), perhaps both before and after training. For the various models trained with different number of scales. The x axis should be a log axis incorporating the entire dynamic range of data in the training set (E-4 to E4 if I understood correctly)

**On instance-normalization:** A simple way to avoid losing information in instance-normalization is to include the information about the sample (mean and std) in an extra token (e.g. [CLS] [SCALE] etc). In the reviewer's experience, this method is the go-to for dealing with samples that have widely varying ranges. I would suggest that the authors perform this comparison. This would make their paper more useful and convincing for a wider audience.

**On forecasting:** The elephant in the room. As forecasting is an important task and desired feature of timeseries foundation models, I believe a clear discussion as to the challenges of decoding in a high dynamic range dataset would make the paper more useful. It is the experience of the reviewer that increasing dynamic range in the output of a neural network is much more difficult than increasing the dynamic range of the input. I would even suggest including any thoughts/experiments in this direction that did not work, as they would be useful for your audience. Since the predominant mode of pre-training on sequences today is autoregressive or masked modeling, the question of decoding will be present in the minds of a large fraction of your readers.

**Fine Tuning for 100 epochs**: I am a little surprised that the finetuning was done for 100 epochs, same as training. Was this necessary? Can you provide some loss curves or some other indication as to why it was done for so long?

Others:
- The authors mention a number of times that they enumerate all possible scales and then mention that this allows them to better generalize to other domains where the scales might be different from the training set. Since all possible scales is an infinite set, I believe the authors mean all scales in the training dataset. As such, the proposed method is still subject to distribution shifts. A test dataset might take values in scales not seen in the training set. I strongly suggest that the authors fix this statement and tone down their claim.

- In the abstract, I suggest changing to something like "To embed scalar values - that my possess arbitrary numerical amplitudes - in a high-dimensional space" for better readability.

- Epochs: For training on large datasets, I think it is  preferrable to mention the total number of tokens (or timesteps) along with the number of epochs. This would make it easier to compare how much a network has been trained in papers that use datasets of different size. It will also make it clear whether the 100 epochs of the training are really the same amount of training as the 100 epochs of fine-tuning.

- In implementation details: Specify how you are resizing the sub-sequence to 512. Are you using some kind of spline? For the scheduler, please specify the final value of the learning rate.  For multivariate data, please give more detail as to where exactly the linear layer is added.

**Strengths And Weaknesses:**

The considered problem is an important step in generalized pre-training of any data that has unbounded range. The authors provide a number of good insights and the exposition is clear.

The paper suffers from a number of weaknesses that need to be addressed before publication. See requested changes section.

---

> ### Author Response · Authors · 2024-05-06
> **Response to reviewer P6SZ**
>
> We thank the reviewer for the valuable feedbacks. The requested changes are incorporated to the revised paper, and highlighted in blue.
>
> Below, we respond to the individual concerns.
>
> * On NME
>   -	We appreciate the reviewer efforts for proposing the simplification. In our opinion, the proposed encoding is mostly equivalent to our NME module. To see this, with an additional linear layer following the proposed encoding, the equation will look very similar to our encoding method. Since our module only contains a paralleled number of linear and layer norm layers, we believe this is not heavily over-engineered. Nonetheless, we agree with the reviewer that the normalization operation $\frac{x}{\sqrt{c_1x^2+c_2^2}}$ and enumerating a few scales are the key to our method.
>   -	We thank the reviewer for pointing this out. We have updated the Figure 3 (b).
>   -	We have added an additional Figure 3(c) to visualize the input and output responses for e(x). The embedding models a complex function which reflects multiple scales of variations. Without training, the encoding responses are rather uniform.
> * Instance normalization baseline with extra input scale tokens.
>
> We thank the reviewer for bringing up this baseline. We follow this suggestion, and append the instance mean and the instance std embeddings as two extra tokens into the Transformer. This gives rise to a classification accuracy of 83.30±0.33 and a Macro-F1 score of 79.65±0.45. The extra input scale tokens improve the baseline instance normalization results of acc 79.97±0.54 and Macro-F1 75.47±0.84. However, it still falls behind our approach by a wide margin. This demonstrates the benefits of window-wise normalization over instance-wise normalization. We have added the results and discussions to the experiment section.
>
> * On forecasting.
>
> We thank the reviewer for the great suggestion. We have added some discussions about decoding and potential future directions. We hope that this could be informative to the readers.
>
> * Is it necessary to finetune for 100 epochs.
>
> We do fine-tuning on each individual downstream dataset (128 datasets in UCR archive), which is by themselves rather small. Because the size between these downstream datasets can also vary, choosing a fixed iteration across datasets can be difficult. The model is finetuned for 100 epochs mainly to increase the effective steps for very small datasets. In practice, fine-tuning for each downstream task runs very fast.
> We also want to note that prior work such as previous best approach HC2 takes 1500 epochs of training, and the recent work SimMTM takes 300 epochs of fine-tuning, both of which are significantly longer than our protocol.
>
> * Minor issues.
>   -	Tone down the claim about distribution shift. We agree with the reviewer, and this is fixed.
>   -	We augment the sequence using the Pytorch function `Random Resized Crop`, which resizes the sequence by bilinear interpolation.
>   -	The end learning rate of the used cosine scheduler is zero.
>   -	The pretraining data with a window size of 16 amounts to about 60 million tokens.
>   -	For encoding multi-variate data, an additional linear layer is added after concatenating each univariate embedding. This linear layer is not shown in the figure.
>   -	We have revised the text to make the above issues clear.

---

> > ### Comment · Reviewer_P6SZ · 2024-05-06
> >
> > I thank the authors for their response. They have addressed the majority of my concerns.
> >
> > I find Fig. 3c very interesting as it seems to show a random projection of the input into a high dimensional space with a log scale. This still makes me wonder if the whole module can be simplified to omit the two layers and layer norm and just construct a simple random projection with $\log(x)$ (appropriately modified to also deal with negative numbers.) I agree with the authors that their construction is lightweight and does not suffer a big computational cost compared to this simplified version. However, I believe there is benefit to having a conceptually simpler construction.

---

> > > ### Author Response · Authors · 2024-05-07
> > > **alternative encoding methods**
> > >
> > > We thank the reviewer for the continued discussions. In our opinion, the main technical challenge is to deal with the numerical discrepancy between high variations of scales and normalized encodings without losing information. The function we followed is the layer norm layer. We definitely believe there are other effective and potentially better normalization functions for encoding numerical values. One of the recent baseline we compared is PLE, which is based on piecewise linear functions. However, it does not appear to be as good as our approach. We will continue to work on this problem for a conceptually simpler construction. We thank the reviewer for the advice again!

---

### Review · Reviewer_3K4m · 2024-04-23

**Summary Of Contributions:**

This paper proposes an embedding module for a large number of time series of different scales. The proposed method is based on Transformer and extract features (mean and std, etc.) for each time-window from data. The effectiveness of the proposed method is verified using various datasets.

**Audience:**

Yes

**Claims And Evidence:**

Yes

**Requested Changes:**

1. The novelty of the proposed method needs to be emphasized more or new technical innovations need to be added.
2. More discussion should be added on the details and limitations of the proposed method (see the above questions).

**Strengths And Weaknesses:**

Strengths
1. The proposed module is model agnostic and generic.
2. The motivation for the research is clear and the paper is well written.
3. Evaluation experiments have been conducted thoroughly.

Weaknesses
1. The technical contribution of the proposed method is small.
2. Contains hyperparameters that are difficult to determine.

Detailed comments:
The proposed method is practical but technically incremental. The proposed method is based on the simple idea of computing the representation at each time-window for multiple scales and applying it to the Transformer. The following are questions.
- How do you optimize the time window and scale candidates? These may have different optimal values for each dataset.
- Can the proposed method be applied to data with different time lengths and time scales?
- The response values may take various forms, not only scale. For example, discrete values or categorical features. Is it possible to extend to such cases?

---

> ### Author Response · Authors · 2024-05-06
> **Response to reviewer 3K4m**
>
> We thank the reviewer for the valuable feedbacks. The requested changes are incorporated to the revised paper, and highlighted in blue.
>
> Below, we respond to the individual concerns.
>
> * On technical novelty.
>
> We hope to re-state and re-emphasize the technical contributions and significance of our work. While there is a convergence of using Transformers for multi-modal AI, data encoding for time series values faces unique challenges unmet by language or vision data. Our core contribution, numerically multi-scaled embedding (NME), successfully addresses the challenge of embedding values with high variations of scales. This simple yet novel contribution enables large-scale learning of time-series data using general purpose Transformer models. Importantly, we demonstrate that our method outperforms heavily engineered hand-crafted features for the first time for deep learning. We have revised the main text to emphasize the significance of the technical contribution as suggested.
>
> * How to choose time window and scale candidate parameters.
>
> The window size is set empirically, and its value is carefully ablated in Table 6. The scale candidate is chosen to cover the mean and the std statistics variations in the training dataset.
> We fully agree with the reviewer that the optimal values of these hyper-parameters may depend on the dataset. Advanced techniques to automatically adjust the hyper-parameters to each dataset may further improve the overall performance. We have added a discussion regarding this issue in the paper.
>
> * Can the method be applied to different time length and time scales.
>
> Yes, due to the nature of the Transformer architecture, our model can process time series signals of arbitrary timesteps. However, as a common practice for batch-mode training, all time-series data is resized to a fixed length of 512 during training.
>
> * Can the method extend to discrete values and categorical features.
>
> We wish to note that discrete values or categorical features can be simply embedded by dictionary lookup tables such linguistic word tokens. They do not share the same numerical encoding challenges as time series values.

---

### Decision · Action_Editor_oMsR · 2024-06-13

**Recommendation:** Accept with minor revision

**Comment:**

The paper was reviewed by three reviewers. After rebuttal, two reviewers were positive (accept and leaning accept), and one reviewer was negative (leaning reject). The negative reviewer cited lack of technical novelty, but acknowledged that the paper "may be practically useful, so it might be accepted." Based on the guidelines of TMLR, I think the paper can be accepted despite its apparent lack of novelty. I recommend "Accept with minor revision" because I hope the authors can release their code as suggested by Reviewer P6SZ.

The reviewers initially had concerns about hyperparameter tuning, ablation studies, forecasting tasks, finetuning, etc. These concerns were addressed by the rebuttal.

**Audience:**

The paper should be of interest to researchers working on pre-training large models for big time series data.

**Claims And Evidence:**

This paper studies the problem of self-supervised pretraining of transformer model on large number (e.g., millions) of time series sequences. The proposed method partitions each sequence into windows. Each window is normalized by the mean and standard deviation. The mean and standard deviation are then embedded into high-dimensional vectors by a numerically multi-scaled embedding module enumerating all possible numerical scales. The proposed method is shown to be effective on a number of tasks. The claims made in the paper are supported by the experiments.